# TITAN: A Trajectory-Informed Technique for Adaptive Parameter Freezing in Large-Scale VQE

**Yifeng Peng[1,†], Xinyi Li[1], Samuel Yen-Chi Chen[2], Kaining Zhang[3], Zhiding Liang[4], Ying Wang[1,*], Yuxuan Du[3,‡]**

[1] Stevens Institute of Technology     [2] Wells Fargo     [3] Nanyang Technological University
[4] Rensselaer Polytechnic Institute

ypeng21@stevens.edu, xli215@stevens.edu, ycchen1989@ieee.org,
kaining.zhang@ntu.edu.sg, zlianghahaha@gmail.com, ywang6@stevens.edu,
yuxuan.du@ntu.edu.sg

## Abstract

Variational quantum Eigensolver (VQE) is a leading candidate for harnessing quantum computers to advance quantum chemistry and materials simulations, yet its training efficiency deteriorates rapidly for large Hamiltonians. Two issues underlie this bottleneck: (i) the no-cloning theorem imposes a linear growth in circuit evaluations with the number of parameters per gradient step; and (ii) deeper circuits encounter barren plateaus (BPs), leading to exponentially increasing measurement overheads. To address these challenges, here we propose a deep learning framework, dubbed TITAN, which identifies and freezes inactive parameters of a given ansätze at initialization for a specific class of Hamiltonians, reducing the optimization overhead without sacrificing accuracy. The motivation of TITAN starts with our empirical findings that a subset of parameters consistently has negligible influence on training dynamics. Its design combines a theoretically grounded data construction strategy, ensuring each training example is informative and BP-resilient, with an adaptive neural architecture that generalizes across ansätze of varying sizes. Across benchmark transverse-field Ising models, Heisenberg models, and multiple molecule systems up to 30 qubits, TITAN achieves up to $3\times$ faster convergence and $40$–$60\%$ fewer circuit evaluations than state-of-the-art baselines, while matching or surpassing their estimation accuracy. By proactively trimming parameter space, TITAN lowers hardware demands and offers a scalable path toward utilizing VQE to advance practical quantum chemistry and materials science.

## 1 Introduction

Quantum computing [1, 2] is widely believed to provide computational advantages in electronic-structure calculations (ESCs) [3], a pivotal task in catalyst discovery [4], drug design [5], and materials innovation [6–9]. The flagship protocol that leverages the power of quantum computers to advance ESCs is the variational quantum Eigensolver (VQE) [10–12], which estimates molecular ground-state energies by iteratively adjusting a parameterized ansätze to lower the expectation value of a fermionic Hamiltonian mapped onto qubits. Recent experiments have verified the potential of VQE in solving ESCs for small-scale systems [13–16]. Despite the progress, scaling VQE from pedagogical circuits to chemically relevant systems confronts two intertwined training bottlenecks, i.e., (i) *barren plateaus*

---

    * † ‡ Corresponding Author. The views expressed in this article are those of the authors and do not represent the views of Wells Fargo. This article is for informational purposes only. Nothing contained in this article should be construed as investment advice. Wells Fargo makes no express or implied warranties and expressly disclaims all legal, tax, and accounting implications related to this article. Appendix is available at
https://github.com/Yifengml/TITAN-Appendix

(BPs) [17, 18] and (ii) expensive *measurement overhead*. BPs describe the exponential suppression of gradient magnitudes in the cost-function landscape as circuit depth or system size grows, thereby impeding effective optimization [19–21]. Over the past years, huge efforts have been devoted to overcoming BPs, where representative works include developing initialization heuristics [18, 22, 23], layerwise training [24, 25], architecture design [26], and adaptive freezing [27–29].

In contrast to the extensive efforts focused on mitigating BPs, the equally fundamental challenge of measurement overhead has received comparatively less attention [30, 31]. This substantial overhead arises from the fundamental constraint that quantum states cannot be cloned and only unitary evolution is permitted, necessitating external reconstruction of analytic derivatives [32]. In particular, the widely used parameter-shift rule requires two circuit evaluations per tunable gate angle, effectively doubling the number of shots per parameter [33–36]. As a result, the total measurement cost scales linearly with the number of parameters and increases with both circuit depth and qubit count. For instance, the benzene ($C_6H_6$), a simple molecular Hamiltonian, requires $10^6$–$10^8$ circuit evaluations (about 5 hours for Ion-trap quantum computers), even after grouping optimizations [37]. Thus, even in the absence of BPs, the scalability of VQE is fundamentally constrained by measurement overhead. Addressing this often-overlooked bottleneck is essential for translating the theoretical potential of VQE into practical quantum-computing workflows.

Initial attempts have been devoted to reducing the measurement overhead in large-scale VQE. Prior strategies fall into several broad categories. Commuting-set partitioning and classical-shadow/derandomization schemes compress the number of shots required to estimate expectation values [38, 39]. Complementary Pauli-term grouping heuristics that exploit qubit-wise commutativity further reduce the sample count in VQEs [40–42]. Quantum architecture search [43–45], QuACK [46], and meta-learning for parameter initialization [47] are proposed to enhance the VQE performance and reduce the overhead. While these complementary approaches alleviate part of the overall computational burden, prior work typically treats them in isolation without a unified perspective. A critical open question remains: can jointly addressing multiple aspects of the VQE optimization pipeline lead to further reductions in measurement overhead and overall computational cost?

We address this question from a new perspective: connecting optimization dynamics with the parameter initialization. This insight stems from our empirical discovery of a "frozen-parameter" phenomenon across diverse Hamiltonians, as shown in Figure 1. This constitutes our first contribution, which may inspire further investigations into parameter dynamics within VQE. Building on this observation, we devise a deep learning (DL) model, TITAN, aiming to harness the power of neural networks to capture and pre-

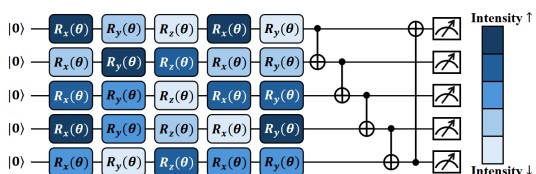

Figure 1: An illustration of the "frozen-parameter" phenomenon of VQE. Intensity refers to the number of inactive sessions. More details are offered in Sec. 4.

dict the frozen parameters of VQE for a given family of Hamiltonians across varying scales. As in other deep learning applications, TITAN comprises three stages: dataset construction, model implementation and training, and model inference.

Despite its promise, implementing TITAN poses notable challenges, especially for dataset curation and model architecture design. During the data construction phase, we proposed the *Adaptive Parameter Freezing and Activation* (APFA) to collect trajectories of inactive parameters from VQE instantiated on diverse Hamiltonians within the target family. A key requirement is that each trajectory must avoid BPs and convey informative training signals. To meet this criterion, we develop a theoretical framework based on Gaussian initialization (see Theorem 1), exhibiting that our data-generation protocol circumvents BP artifacts. This theoretical result forms a core technical contribution of our work and may be of independent interest. For the model implementation, we introduce the *coordinate-aware fully-convolutional self-attention* (CFCSA) technique to enhance TITAN's adaptability, enabling it to accommodate variations in qubit count, circuit depth, and Hamiltonian structure without retraining. Our final contribution is a systematic evaluation of TITAN's performance through extensive experiments on standard lattice and molecular systems with up to 30 qubits. The achieved results exhibit that TITAN attains $\geq 99\%$ of the ground-state energy reached by the baseline model by freezing at least $40\%$ of parameters.

## 2 Background and Related Works

**Quantum Computing** Quantum computing leverages intrinsically quantum-mechanical phenomena—superposition, entanglement, and interference [48, 49]. In this framework, the fundamental unit of information is the *qubit*, a two-level quantum system that can occupy any complex linear combination of computational basis states, i.e., $|\psi\rangle = \alpha|0\rangle + \beta|1\rangle$, with $\alpha, \beta \in \mathbb{C}$ and $|\alpha|^2 + |\beta|^2 = 1$. An $n$-qubit state is represented by a vector in a $2^n$-dimensional Hilbert space, $|\Psi\rangle = \sum_{x=0}^{2^n-1} c_x |x\rangle$, $\sum_{x=0}^{2^n-1} |c_x|^2 = 1$. Here $|x\rangle$ denotes the computational basis for $n$ qubits, where $|x_1 x_2 \ldots x_n\rangle$ corresponds to a binary string $(x_1, x_2, \ldots, x_n)$ of length $n$, and is defined by the tensor-product rule $|x\rangle = |x_1\rangle \otimes |x_2\rangle \otimes \ldots \otimes |x_n\rangle$.

Quantum gates amount to the unitary operations that transform one quantum state into another. *Solovay–Kitaev theorem* for standard constructions [1, 50], which guarantees that any unitary operation can be decomposed into single and two-qubit gates. Measurement in quantum mechanics is described by a *positive operator–valued measure* (POVM) $\{M_m\}$ that satisfies the completeness relation $\sum_m M_m^\dagger M_m = I$. In the special—but ubiquitous case of a *projective measurement* in the computational basis, the POVM elements reduce to the projectors $M_x = |x\rangle\langle x|$, $x \in \{0,1\}^n$, so that measuring an $n$-qubit state $|\Psi\rangle = \sum_{x=0}^{2^n-1} c_x |x\rangle$ yields outcome $x$ with probability $p_x = |\langle x|\Psi\rangle|^2 = |c_x|^2$, after which the post-measurement state collapses to $|x\rangle$.

**Variational Quantum Eigensolver (VQE)** In quantum chemistry [51–54], one is often interested in finding the ground-state energy of a molecular system described by the Hamiltonian: $\hat{H} = \sum_i h_i \hat{H}_i$, where each $\hat{H}_i$ is a tensor product of Pauli operators $I = \left(\begin{smallmatrix} 1 & 0 \\ 0 & 1 \end{smallmatrix}\right)$, $X = \left(\begin{smallmatrix} 0 & 1 \\ 1 & 0 \end{smallmatrix}\right)$, $Y = \left(\begin{smallmatrix} 0 & -i \\ i & 0 \end{smallmatrix}\right)$, $Z = \left(\begin{smallmatrix} 1 & 0 \\ 0 & -1 \end{smallmatrix}\right)$, and the real coefficients $h_i$ encode system-specific molecular parameters [55, 56]. The VQE aims to approximate the ground state $|\psi_{\mathrm{gs}}\rangle$ of $\hat{H}$ by employing a parameterized quantum circuit (ansätze) $|\psi(\boldsymbol{\theta})\rangle = U(\boldsymbol{\theta})|\psi_0\rangle$, where $|\psi_0\rangle$ is an easily preparable reference state (e.g., the Hartree–Fock state in chemistry), and $U(\boldsymbol{\theta})$ is a unitary operator constructed from quantum gates whose adjustable parameters $\boldsymbol{\theta} = \{\theta^{(1)}, \theta^{(2)}, \ldots \theta^{(P)}\}$ are optimized to minimize the energy expectation value. The generic form of an $N$-qubit ansätze is

$$U(\boldsymbol{\theta}) = \prod_{i=1}^{p} \left( W_i V_i(\theta^{(i)}) \right), \tag{1}$$

where $V_i(\theta^{(i)})$ denotes the $i$-th tunable gate and $W_i$ refers to the $i$-th fixed unitary operation. Existing ansätze used in VQEs can be classified into two classes: *structured* ansätze and *unstructured* ansätze. For *structured* ansätze, they are commonly designed with a layer-wise structure, comprising repeated blocks of parameterized gates and entangling gates. A typical example in this class is the hardware-efficient ansatz (HEA) [57], e.g., $U(\boldsymbol{\theta}) = \prod_{\ell=1}^{L} (W_\ell \otimes_{i=1}^{N} \mathrm{RX}(\theta^{(i,\ell)}))$ with $\mathrm{RX} = \exp(-i\theta X/2)$, $W_\ell = \prod_{i=1}^{N-1} \mathrm{CNOT}_{i,i+1}$, and $\mathrm{CNOT}_{i,i+1}$ being applying $\mathrm{CNOT} = |0\rangle\langle 0| \otimes I + |1\rangle\langle 1| \otimes X$ gate to the $i$-th and $i+1$-th qubits. Other notable structured ansätze include SU2 [58, 59] and SEL [60] ansätze. In what follows, we denote an $N$-qubit VQE with a structured ansätze as $(\hat{H}, \mathcal{A}(L, N, D))$, where $D$ denotes the number of variational parameters per qubit wire in each block and $p = L \times D \times N$. For *unstructured* ansätze, they generally encode prior knowledge of the explored Hamiltonian in $U(\boldsymbol{\theta})$, where the corresponding gate layout is denoted by $\mathcal{S}$. Typical instances in this regime include unitary coupled cluster (UCC) [61] and unitary coupled cluster with singles and doubles (UCCSD). We denote an $N$-qubit VQE with a *unstructured* ansatz as $(\hat{H}, \mathcal{A}(p, \mathcal{S}))$.

The objective of VQE is to approximate ground-state energy $E_0$ of $\hat{H}$ by optimizing $\boldsymbol{\theta}$, i.e.,

$$E(\boldsymbol{\theta}) = \langle \psi(\boldsymbol{\theta})|\hat{H}|\psi(\boldsymbol{\theta})\rangle \longrightarrow E_{\min} = \min_{\boldsymbol{\theta}} E(\boldsymbol{\theta}) \approx E_0. \tag{2}$$

The optimization is often completed by the gradient-descent optimizer, where the update rule yields $\theta_j \leftarrow \theta_j - \eta \, \partial E / \partial \theta_j$, and $\eta$ is the learning rate [19, 20].

**Related works** Improving the optimization efficiency of large-scale VQE remains a central challenge in quantum computing [62, 63]. Recent progress falls into three major directions: *(i) Measurement grouping.* Learning-based grouping [64] and classical-shadow protocols [65, 66] compress the number of shots required to estimate large Hamiltonians, but may incur substantial classical post-processing overhead for high-qubit systems. *(ii) ansätze design.* Hardware-efficient circuits [57]

suffer from BPs; remedies include quantum architecture search [43, 67], pruning [27, 68], and domain-informed ansätze [69]. These improve expressibility and trainability at the expense of costly search procedures and domain expertise. *(iii) Advanced optimizers.* Quantum-aware optimizers such as *Quack* [46] and meta-learning frameworks [70, 71, 47], non-convex landscapes, but introduce additional hyperparameters and nested optimization loops. Collectively, these three lines of research are *orthogonal* to our contribution. TITAN addresses a **fourth** axis—*parameter–space reduction via predictive freezing*—that can be layered *on top of* any measurement-grouping scheme, circuit-design strategy, or optimizer, thereby providing a complementary boost to existing categories. More details of related works are in Appendix A.

## 3 "Frozen-parameter" Phenomenon and Implementation of TITAN

In this section, we first present APFA scheme, using to exhibit the "frozen-parameter" phenomenon in VQE. We then show the implementation details of the proposed TITAN.

### 3.1 Phenomenon of Frozen Parameters in VQE

VQE can exhibit a parameter–freezing effect in which portions of the variational parameters become static during optimization; however, the extent of this phenomenon has never been quantified. Here we introduce the APFA—to our knowledge, the first framework that records the full, time-resolved mask trajectory of every parameter, thereby providing concrete numerical evidence of how, when, and to what degree individual angles freeze over the course of training.

**APFA** The APFA mechanism dynamically identifies and *freezes* low-saliency parameters in a VQE (at the $t$-th iteration) to record the freezing trajectory by executing the following four steps.

(Step i) For every coordinate $i$, keep an exponential moving average (EMA) [72] of the *absolute* gradient $\widehat{g}_t^{(i)} = \alpha \widehat{g}_{t-1}^{(i)} + (1-\alpha)\big|g_t^{(i)}\big|, 0 < \alpha < 1$, where $g_t^{(i)}$ is the instantaneous gradient $\partial f/\partial \theta^{(i)}$ at iteration $t$, and $\alpha$ is a smoothing factor controlling memory depth.

(Step ii) For a VQE with parameters $\boldsymbol{\theta}_t$ at iteration $t$, we define the stochastic gradient $\mathbf{g}_t = \nabla_{\boldsymbol{\theta}} f(\boldsymbol{\theta}_t) + \boldsymbol{\xi}_t$, where $\boldsymbol{\xi}_t$ is an isotropic noise term sampled from a multivariate normal distribution $\mathcal{N}(\mathbf{0}, \gamma^2 \mathbf{I})$ to improve exploration. Let $r_t = \|\mathbf{g}_t\|_2/(\|\mathbf{g}_0\|_2 + \varepsilon)$ be the *gradient-decay ratio* with a small constant $\varepsilon$. Two scale factors, $\lambda_f^{(t)}$ (freeze) and $\lambda_a^{(t)}$ (activate), are modulated by $r_t$,

$$
\begin{aligned}
\lambda_f^{(t)} &= \lambda_{f,\min} + (1-r_t)(\lambda_{f,\max} - \lambda_{f,\min}),\ 0 < \lambda_{f,\min} < \lambda_{f,\max} \leq 1 \\
\lambda_a^{(t)} &= \lambda_{a,\min} + (1-r_t)(\lambda_{a,\max} - \lambda_{a,\min}),\ 1 < \lambda_{a,\min} < \lambda_{a,\max}
\end{aligned}
\tag{3}
$$

where $\lambda_{f,\min/\max}$ and $\lambda_{a,\min/\max}$ are four hyperparameter. Then, let $\bar{g}_t = \frac{1}{P}\sum_{i=1}^{P} \widehat{g}_t^{(i)}$ be the mean EMA magnitude across $p$ parameters. Following this and Eq. (3), the *freeze* and *activate* thresholds for APFA at the $t$-th iteration are $\tau_f^{(t)} = \lambda_f^{(t)}\bar{g}_t$ and $\tau_a^{(t)} = \lambda_a^{(t)}\tau_f^{(t)}$.

(Step iii) Each parameter maintains two counters: $c_f^{(i)}$, i.e., the number of successive optimization iterations with $\widehat{g}_t^{(i)} < \tau_f^{(t)}$; and $c_a^{(i)}$, i.e., the number of successive iterations $\widehat{g}_t^{(i)} > \tau_a^{(t)}$. These counters together form a binary *mask* $m_t^{(i)} \in \{0,1\}$ indicating its state with 1 being active and 0 being frozen, respectively. A parameter is **frozen** when $c_f^{(i)} \geq N_f$ and **reactivated** when $c_a^{(i)} \geq N_a$, where $N_f$ and $N_a$ are hyperparameter referring to the patience lengths.

(Step iv) Denote the mask vector as $\mathbf{m}_t$ and the learning rate at the $t$-th iteration as $\eta_t$. The first-order optimizer performs the *Hadamard-masked* update $\boldsymbol{\theta}_{t+1} = \boldsymbol{\theta}_t - \eta_t(\mathbf{m}_t \odot \mathbf{g}_t)$, where $\odot$ is the Hadamard product. As such, only active parameters are updated.

After executing the four APFA steps for $T$ iterations, we stack the iteration-wise binary masks into $\mathbf{Y} = \{\mathbf{m}_t\}_{t=0}^{T} \in \{0,1\}^{(T+1)\times P}$, where each row $\mathbf{m}_t$ indicates, at iteration $t$, which of the $P$ trainable parameters are frozen (0) or active (1). The cumulative sum of this sequence, $\mathbf{C} = \sum_{t=0}^{T} \mathbf{m}_t \in \mathbb{R}^P$, serves as a compact "intensity" measure of how often each parameter has been frozen during the entire training process (see Appendix B for further details).

**Observation of frozen parameters** Based on the proposed AFPA, we report the *cumulative freezing counts* of all parameters of HEA when applied to estimate the ground state energy of the isotropic Heisenberg Hamiltonians. By varying the qubit size from 0 to 15 qubits, and ranging the layer number of HEA from 0 to 15, the achieved results are shown in Figure 2. The color map ranges from white to dark blue (frozen in nearly every epoch). Empirical evidence shows that initialization decisively shapes the subsequent saliency landscape: parameters that start in poorly conditioned regions are rapidly driven to negligible gradients, whereas well-scaled directions remain trainable throughout the run. This tight coupling between initialization and training dynamics suggests that early-stage geometry can predetermine

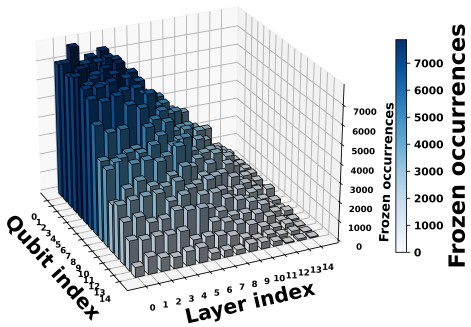

Figure 2: Statistics of the frozen parameters intensity in HEA optimized for isotropic Heisenberg Hamiltonian.

long-term redundancy patterns. Hence, the optimization trajectory naturally partitions the parameter space into *persistently active* and *repeatedly redundant* dimensions. This phenomenon delivers two key insights that connect the parameter initialization and the optimization dynamics. First, parameters identified as inactive can be frozen a priori, reducing the optimization overhead without sacrificing accuracy. Second, it is possible to leverage learning models to predict those inactive parameters.

## 3.2 TITAN: Dataset Construction and Modeling

Motivated by these insights, we propose TITAN. It demonstrates that (i) frozen parameters can be *predicted a priori* and (ii) this knowledge markedly reduces VQE measurement overhead without affecting the accuracy. At a high level, TITAN establishes a deep predictive model to estimate a *freeze-intensity tensor* $\mathbf{Y}$ returned by APFA, given the specified Hamiltonian and ansatz. After training, TITAN can predict frozen parameters given any new ansatz and Hamiltonian to reduce VQE measurement overhead. The implementation of TITAN comprises three stages: dataset construction, model training, and inference. We next describe each stage in detail.

### 3.2.1 Dataset Construction

The training dataset employed in TITAN takes the form as $\mathcal{D} = \{(\mathbf{X}_i, \mathbf{C}_i)\}_{i=1}^{M}$, where $M$ are data points and each pair consists of (i) an input tensor $\mathbf{X}_i$ capturing the relevant features of a circuit configuration, and (ii) the corresponding freeze-intensity tensor $\mathbf{C}_i$, which indicates the activity of each parameter recorded by APFA.

A critical issue in constructing $\mathcal{D}$ is ensuring that each data pair is informative. More precisely, when a VQE instance suffers from BPs, the parameter-freezing trajectory returned by APFA fails to yield useful information. To address this, we refine the original Gaussian initialization scheme [18] to ensure the trainability of large-scale VQE for a broad class of ansätze. This result is formalized in the following theorem, with the corresponding proof provided in Appendix C.

**Theorem 1** (Enhanced Gaussian Initialization). *Let $H$ be the explored $N$-qubit local Hamiltonian. Following notations in Eq.* (1)*, when the employed HEA $U(\boldsymbol{\theta})$ yields $\left(W_{2\ell-1}, V_{2\ell-1}(\boldsymbol{\theta}^{(2\ell-1)})\right) = \left(M_\ell CZ_\ell, RY_\ell(\boldsymbol{\theta}^{(2\ell-1)})\right)$, $\left(W_{2\ell}, V_{2\ell}(\boldsymbol{\theta}^{(2\ell)})\right) = \left(\mathbb{I}, RY_\ell(\boldsymbol{\theta}^{(2\ell)})\right)$, $\forall \ell = 1, \ldots, L$ with $M_\ell = \mathbb{I}$ for $\ell > 1$ and $M_1$ be the tensor product of fixed single-qubit unitary $\{U_n\}_{n=1}^{N}$ independently sampled from $\{R_Z(\pm\frac{\pi}{4}),\ R_Y(\frac{\pi}{4})R_X(\pm\frac{\pi}{4}),\ R_X(\frac{\pi}{4})R_Y(\pm\frac{\pi}{4})\}$.*

*Define the expectation $f(\boldsymbol{\theta}) = \langle 0|U(\boldsymbol{\theta})^\dagger O U(\boldsymbol{\theta})|0\rangle$. Then $\mathbb{E}_{\boldsymbol{\theta}, U_1, \ldots, U_N}[(\partial_{\theta^{(j,n)}} f)^2] \geq \Theta(1/L)$ when each element in $\boldsymbol{\theta} := (\boldsymbol{\theta}^{(1)}, \cdots, \boldsymbol{\theta}^{(2L)})$ is sampled independently from $\mathcal{N}(0, \gamma^2)$ with $\gamma^2 = \mathcal{O}(1/L)$.*

Theorem 1 provides a *non–vanishing* lower bound on the square of partial derivative that scales at worst as $\Theta(1/L)$ for the depth-dependent choice $\gamma^2 = c/L$ with $c > 0$. Therefore, the overall gradient norm remains $\Omega(1)$ rather than decaying as $2^{-\Omega(L)}$. Hence, the parameter space does *not* collapse into an exponentially flat plateau at initialization. Compared to the original Gaussian Initialization [18] that considers Pauli matrix observables, Theorem 1 has milder conditions on the observable format, which suits the VQE for quantum many-body Hamiltonians.

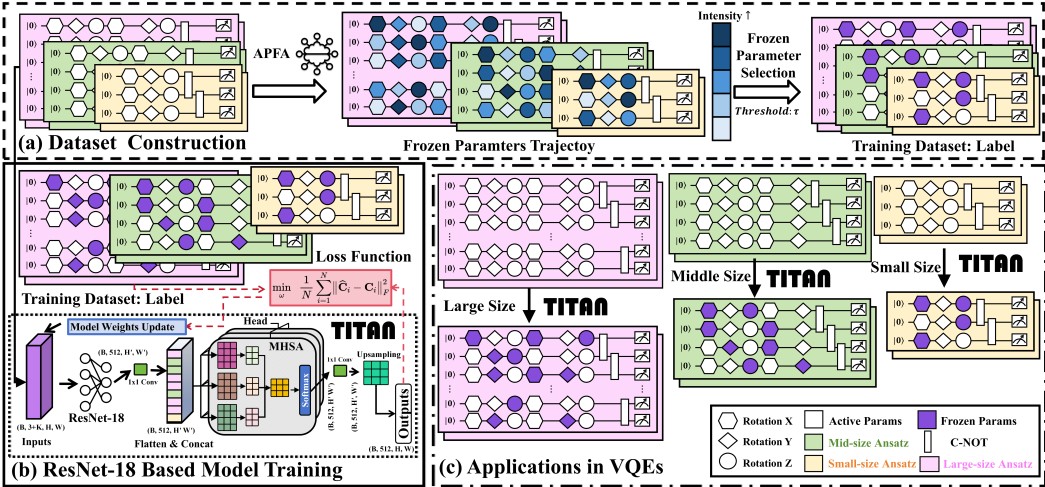

Figure 3: Overview of TITAN framework. TITAN predicts the frozen parameters of different ansätze scales. (a) Dataset construction, (b) Model Training, (c) Extension of the multiscale ansätze.

### 3.2.2 Modeling Implementation and Optimization of TITAN

A deep neural network $h_{\boldsymbol{\omega}}(\cdot)$ is employed to map $\mathbf{X}_i$ to an estimate $\widehat{\mathbf{C}}_i = h_{\boldsymbol{\omega}}(\mathbf{X}_i)$. We employ a ResNet-18 backbone [73] augmented with a multi-head self-attention (MHSA) [74] layer acting *channel-wise* to predict the intensity tensor $\mathbf{C}$. Let $h_{\boldsymbol{\omega}}(\cdot)$ denote this mapping from $\mathbf{X}$ to $\hat{\mathbf{C}}$. To learn $\boldsymbol{\omega}$, one minimizes a loss function over the dataset $\mathcal{D}$: $\min_{\boldsymbol{\omega}} \frac{1}{N} \sum_{i=1}^{N} \left\| \widehat{\mathbf{C}}_i - \mathbf{C}_i \right\|_F^2$, where $\| \cdot \|_F$ denotes the Frobenius norm. Modern optimizers such as Adam are typically used to minimize the loss function, and the training details are elucidated in the Appendix D.

**CFCSA encoding for arbitrary Hamiltonians and ansätze layouts.** The CFCSA (Coordinate + Descriptor Fully Convolutional & Self-Attention) encoding scheme can handle arbitrarily structured $(\hat{H}, \mathcal{A}(L, N, D))$ or unstructured VQE ansätze circuits $(\hat{H}, \mathcal{A}(p, \mathcal{S}))$ introduced in Section 2.

For Coordinate Channels (3D), each parameter is associated with a triple $\left[ \frac{\ell}{L-1}, \frac{d}{D-1}, \frac{q}{N-1} \right]$, where $\ell \in \{0, \ldots, L-1\}$ tracks the layer index, $d \in \{0, \ldots, D-1\}$ tracks the gate index within each layer, and $q \in \{0, \ldots, N-1\}$ tracks which qubit is targeted. These three normalized coordinates form the first three channels, allowing the network to recognize positional and structural roles of each parameter (*e.g.*, early vs. late layer, different rotation axes, distinct qubits).

For Descriptor Channels ($K$-D), a set of external scalars $\mathcal{S} = \{s_1, \ldots, s_K\}$ provides problem-specific context (e.g. coupling constants, symmetry tags, or UCC-specific hyperparameter). Each descriptor $s_k$ is normalized to $[0, 1]$ and then broadcast over the entire $p$ grid, creating $K$ additional channels. Hence, changes in the Hamiltonian $\hat{H}(\mathcal{S})$ or ansätze metadata are passed directly into the network as an extra $(K)$-dimensional *descriptor* input.

After stacking all channels, we obtain an input tensor: $\mathbf{X} \in \mathbb{R}^{(3+K) \times L \times (DN)}$. This tensor is processed by: *Convolutional Layers (stride = 1)* that preserve spatial dimensions, and then *MHSA* over the flattened $(L \times DN)$ tokens. The key insight is that both the convolutional filters (stride 1) and the global MHSA mechanism are *dimension-agnostic*, allowing the same backbone weights $\boldsymbol{\omega}$ to process any $(L, N, D)$ combination. Thus, no architectural modifications are needed to handle different depths, qubit counts, or gate sets.

### 3.2.3 Inference Phase and Complementary to Other Methods

TITAN can *generalize* across unseen Hamiltonians: changing $\mathcal{S}$ modifies descriptor channels, injecting new context without altering the network structure. *Arbitrary Circuit Architectures* simply changes the input shape, which the fully convolutional + MHSA backbone can still process. Hence, CFCSA of TITAN enables a single predictor to operate on vastly different VQE setups, offering

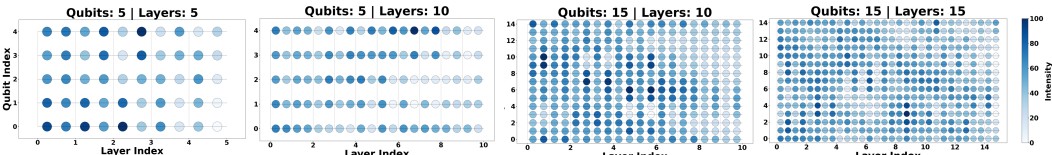

Figure 4: Frozen parameters intensity under APFA. Each panel shows the intensity (darker refers to more often frozen) when applying VQE with HEA to the isotropic Hamiltonian.

Table 1: $\Delta E$ under Gaussian initialization. Frozen threshold ($\tau = 80$) is marked with $*$ ; $\tau = 90$ is not. $\Delta E \leq 0$ is highlighted with green , frozen parameters $\geq 5$ is colored with cyan .

| | N: **5** | N: **6** | N: **7** | N: **8** | N: **9** | N: **10** | N: **11** | N: **12** | N: **13** | N: **14** | N: **15** |
|---|---|---|---|---|---|---|---|---|---|---|---|
| **Proposed TITAN: Final Energy Comparison ($\Delta E = E_{\text{Titan Gauss}} - E_{\text{Baseline Gauss}}$)** | | | | | | | | | | | |
| L: 5 | −0.104 | +0.156 | +0.324 | −0.068 | −0.092* | −0.050 | +0.582 | −0.366 | +0.434 | −0.350 | −0.315 |
| L: 6 | −0.084* | −0.069 | −0.138 | −0.033 | −0.014* | +0.354 | −0.181 | +0.217 | −0.024 | −0.157 | −0.087 |
| L: 7 | −0.222* | −0.025* | −0.035 | −0.136 | −0.054 | +0.282 | −0.002* | −0.008* | −0.141 | −0.140 | +0.538 |
| L: 8 | −0.015 | −0.022* | +0.176 | +0.147 | +0.146 | −0.127 | −0.029* | −0.024 | +0.184 | −0.077 | +0.426 |
| L: 9 | −0.047 | −0.024* | −0.114* | −0.030* | −0.119 | −0.120* | −0.122 | −0.114* | −0.177* | −0.043 | −0.063* |
| L: 10 | −0.084* | −0.025 | −0.136 | −0.022 | −0.011 | +0.131 | −0.033* | −0.018 | −0.003 | −0.042 | −0.074 |
| **Number of Frozen Parameters** | | | | | | | | | | | |
| L: 5 | 3/50 | 3/60 | 3/70 | 2/80 | 2/90 | 1/100 | 29/110* | 2/120 | 2/130 | 2/140 | 8/150 |
| L: 6 | 5/60* | 1/72 | 1/84 | 4/96 | 1/108* | 4/120 | 5/132 | 3/144 | 5/156 | 3/168 | 7/180 |
| L: 7 | 9/70* | 15/84* | 2/98 | 2/112 | 3/126 | 3/140 | 14/154* | 6/168* | 6/182 | 12/196 | 4/210 |
| L: 8 | 1/80 | 8/96* | 5/112 | 4/128 | 1/144 | 5/160 | 4/176* | 9/192 | 2/208 | 5/224 | 1/240 |
| L: 9 | 4/90 | 10/108* | 9/126* | 7/144* | 2/162 | 11/180* | 1/198 | 26/216* | 14/234* | 2/252 | 15/270* |
| L: 10 | 7/100* | 1/120 | 1/140 | 2/160 | 2/180 | 11/200 | 11/220* | 1/240 | 2/260 | 5/280 | 1/300 |

scalability and transferability across the full design space of $(\hat{H}, \mathcal{A}(L, N, D))$ or $(\hat{H}, \mathcal{A}(p, \mathcal{S}))$ pairs. In Section 4 and Appendix E, we show TITAN is complementary to other methods.

# 4 Experiments

Having established the validity of our training corpus, we now detail both the dataset–generation experiments and the subsequent validation protocol for TITAN. The empirical study is organized into two complementary suites: Heisenberg spin models and Quantum-chemistry benchmarks.

**Hamiltonian with Varying Qubits and Layers** We first consider a Heisenberg Hamiltonian $H = \sum_{i=1}^{N-1}(a\,X_iX_{i+1} + b\,Y_iY_{i+1} + c\,Z_iZ_{i+1})$, where $N \in \{5, \ldots, 15\}$ is the number of qubits. The coefficients are set as $(a, b, c) \in [-5, 5]^3$.

**Quantum-chemistry benchmarks** After Jordan–Wigner or Bravyi–Kitaev transformation, the molecular Hamiltonian $\hat{H}$ becomes a weighted sum of Pauli strings. So $|\psi(\boldsymbol{\theta})\rangle = V(\boldsymbol{\theta}) |\phi_{\text{HF}}\rangle$, the energy function in Eq. (2) becomes $E(\boldsymbol{\theta}) = \langle\phi_{\text{HF}}| V^\dagger(\boldsymbol{\theta})\,\hat{H}\,V(\boldsymbol{\theta}) |\phi_{\text{HF}}\rangle$.

**Metric** Throughout this work, we evaluate the performance of *both* Heisenberg Hamiltonian and quantum chemistry by the variational energy itself as Eq. (2):

$$\text{lower } E(\boldsymbol{\theta}) \implies \text{better approximation of the exact ground state.} \quad (4)$$

**Settings** All experiments are implemented by Tencent Quantum Tensor Circuits [75] and Pennylane [76] with NVIDIA GeForce RTX 3060. More setting details are introduced in the Appendix F.

**"Frozen-parameter" Phenomenon** We first exhibit that APFA enables a stable and interpretable sparsity pattern that persists across circuit scales as shown in Figure 4 when applying VQE with HEA to isotropic Hamiltonian with $L \in \{5, \ldots, 10\}$ and $N \in \{5, \ldots, 15\}$. For $5 \times 5$ circuits, APFA freezes almost all angles in the first two layers. Increasing the depth to $5 \times 10$ shifts the active region to the circuit tail, concentrating optimization near the output. With 15 qubits, there are also significantly more activation parameters on the output side.

**HEA (Isotropic Hamiltonian)** We employ a HEA with $L$ layers and $N$ qubits, $L \in \{5, \ldots, 10\}$ and $N \in \{5, \ldots, 15\}$. In the upper block of the Table 1, more than $90\%$ of the cells are shaded

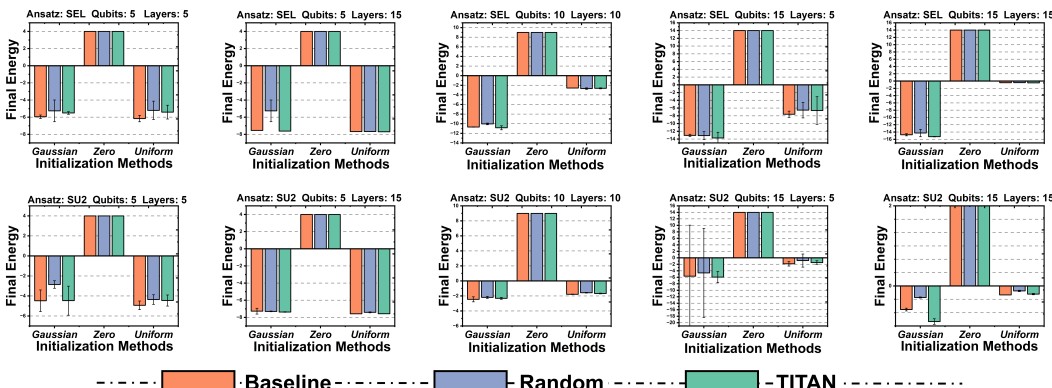

Figure 5: Comparison of the final energies obtained using different initialization methods (Gaussian [18], Zero, Uniform, and TITAN) and optimization strategies (Baseline (Vanilla), Random Freeze, TITAN) with gradient descent optimizer. The results are shown for the isotropic Hamiltonian with SEL (**upper row**) and SU2 (**lower row**), under the threshold $\tau = 80$. Lower final energy values indicate better optimization performance. Frozen parameters refer to the Appendix F.

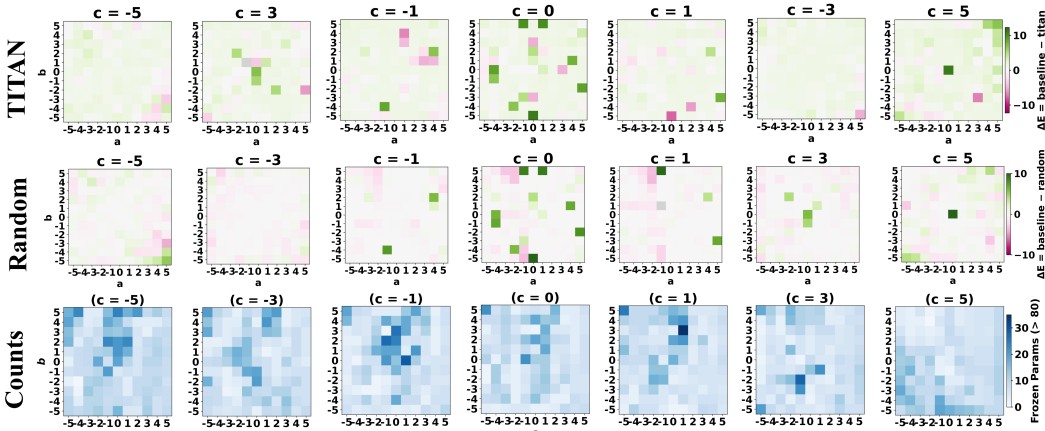

Figure 6: Energy difference heat-maps for the HEA isotropic Heisenberg Hamiltonian with Gaussian initialization. Each panel displays $\Delta E(a, b, c) = E_{\text{baseline}} - E_{\text{init}}$ on a coefficient grid with horizontal axis $a \in [-5, 5]$ and vertical axis $b \in [-5, 5]$; the seven columns sweep the $ZZ$ coupling coefficient $c \in -5, -3, -1, 0, 1, 3, 5$ from left to right. **Top row**: TITAN. **Middle row**: *Random Freezing*. **Bottom row**: Froze parameters counts ($\tau = 80$). Positive (green) values indicate that the baseline run converges to a higher final energy than the compared strategy, whereas negative (magenta) values signify energy losses relative to the baseline. The color scale is clipped symmetrically about zero.

green ($\Delta E \leq 0$), indicating that TITAN's data-driven mask either matches or surpasses the baseline across almost every $\langle$layers, qubits$\rangle$ setting where **Baseline** is without any parameters frozen. And almost $50\%$ of the cells are shaded cyan . Therefore, TITAN can also adjust the number of frozen parameters by adjusting the threshold $\tau$ to achieve the energy we need that is relatively lower or similar to the baseline ($\Delta E \leq 0$). Additional threshold results appear in the Appendix F.

**Test in SU2 and SEL (Isotropic Hamiltonian)** On the other hand, to prove the generalization of TITAN, we also tested it on other ansätze, i.e., SU2 and SEL, as shown in the Figure 5. We show how TITAN complements traditional initialization methods, such as Zero initialization, uniform initialization, and traditional Gaussian initialization [18]. Random Freeze never outperforms Baseline and often incurs energy penalties of 0.3–1.2 Ha. This underscores that the benefit comes from pruning targeted rather than a simple reduction. The success of TITAN on unseen ansätze and Hamiltonians suggests that the learned mask captures Hamiltonian-invariant directions in the parameter landscape.

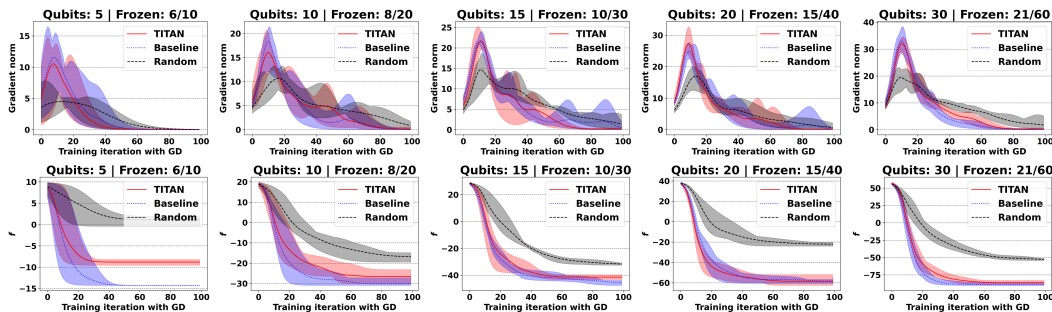

Figure 7: Convergence behavior of TITAN (solid red) versus the *Baseline* (blue dotted) and a *Random-freeze* control (black dashed) TFIM with coupling $J = -3$ and transverse field $h = 2$ with Gaussian initialization [18]. The **top row** plots the $\ell_2$-norm, while the **bottom row** tracks $f$ (final energy). Shaded envelopes denote $\pm\sigma$ across **5 independent runs** with threshold ($\tau = 80$).

Table 2: Energy differences between Molecules with three initialization strategies: Gaussian [18] ($\diamondsuit$), zero ($\heartsuit$), and uniform ($\clubsuit$) under TITAN freezing and Random freezing. Less energy is colored with blue and worse is red with ($\tau = 80$).

| | $H_2$ (4) | HF (4) | LiH (10) | $BeH_2$ (12) | $H_2O$ (10) | $N_2$ (12) | CO (12) |
|---|---|---|---|---|---|---|---|
| | $\Delta E = E_{\text{TITAN}} - E_{\text{Baseline}}$ | | | | | | |
| $\diamondsuit$ | 0.000 | 0.000 | 0.018 | 0.008 | 0.000 | $-0.007$ | $-0.015$ |
| $\heartsuit$ | 0.000 | 0.000 | $-0.017$ | $-0.008$ | 0.000 | 0.007 | 0.015 |
| $\clubsuit$ | 0.497 | $-0.148$ | 0.145 | $-0.236$ | $-0.078$ | 2.512 | $-1.729$ |
| | $\Delta E = E_{\text{Random}} - E_{\text{Baseline}}$ | | | | | | |
| $\diamondsuit$ | 2.248 | 0.000 | 0.003 | 0.013 | 0.004 | 0.011 | 0.017 |
| $\heartsuit$ | 2.248 | 0.001 | 0.003 | 0.013 | 0.004 | 0.010 | 0.017 |
| $\clubsuit$ | 1.991 | 0.573 | $-0.016$ | 0.112 | 0.621 | 1.524 | 1.743 |
| | **Number of Frozen Parameters** | | | | | | |
| | 1/3 | 1/3 | 10/24 | 21/92 | 4/54 | 47/117 | 77/117 |

**HEA (Anisotropic Hamiltonian)** In terms of Anisotropic Hamiltonian, Figure 6 reports the *final–energy gap* $\Delta E(a, b, c) = E_{\text{baseline}} - E_{\text{init}}$ for a 8-qubit, 5-layer HEA. Across the entire $(a, b, c)$ hypersurface TITAN yields predominantly green cells, with improvements reaching $\Delta E \approx 0.12$. Gains are especially pronounced for $|c| \geq 3$ (i.e. strongly $ZZ$-dominated regimes) and for balanced couplings $|a| \approx |b|$, highlighting TITAN's robustness under both isotropic and anisotropic interactions. However, when considering random parameters freezing, it exhibits a near-zero mean: green and magenta cells are interspersed without discernible structure, and the absolute deviations seldom exceed $|\Delta E| < 0.03$. This corroborates that the performance lift stems from TITAN's informed freezing strategy rather than stochastic variance.

For $c = 0$ (pure $XY$ model) TITAN's advantage narrows yet remains positive, whereas for $ZZ$-dominated columns ($c = \pm 5$) improvements are both larger in magnitude and spatially wider. This pattern suggests that circuits with stronger longitudinal entanglement profit more from TI-TAN's early dimensionality reduction. The heat maps are approximately symmetric with respect to $(a, b) \mapsto (-a, -b)$, in line with the underlying Hamiltonian symmetry; TITAN preserves this property, indicating that the predictor does not inject bias towards a particular sign of the couplings.

**TFIM (Anisotropic Hamiltonian)** Compared with both reference baselines, TITAN exhibits markedly superior optimization dynamics across all examined qubit counts. In the *top-row* panels of Figure 7, the $\ell_2$-norm of the gradient for TITAN (solid red) diminishes precipitously within the first $\sim 10$ training iterations, whereas the *Baseline* (blue dotted) and *Random-freeze* control (black dashed) maintains substantially higher gradient amplitudes for 20–40 iterations. This steeper decay implies that TITAN more rapidly enters a region of the parameter manifold where the objective landscape is locally smoother, thereby reducing the variance of subsequent updates. The *bottom-row* panels reveal

a correspondingly accelerated decrease in the objective value $f$ (final variational energy): TITAN attains energies lower than the Baseline after fewer than 15 iterations for $N \leq 15$ qubits and sustains an advantage that widens with system size, reaching a gap of $\sim 1.5$–$2.0$ energy units at $N = 20$. Moreover, the shaded envelopes are markedly narrower for TITAN, indicating reduced run-to-run variability and greater stability.

**Quantum Chemistry Molecules** We also present the molecules experiment in the Table 2. For most molecules (e.g., $H_2$, HF, and LiH), the Gaussian initialization under TITAN yields negligible $\Delta E$, suggesting that these methods closely match or slightly improve upon the baseline. In contrast, Uniform initialization can lead to larger deviations, indicating higher energy. The last row reveals that $30\%$ to $66\%$ of parameters can be frozen *a priori* (e.g. 77/117 (65.8%) for CO) without degrading energy, validating TITAN's performance in VQE measurement overhead reduction.

## 5 Conclusion

This work introduced TITAN, the first end-to-end framework that *learns* to predict frozen-parameter intensity for VQE. By harvesting large-scale frozen parameter trajectories by APFA, we created a labeled dataset that exposes persistent redundancies in VQE ansätze. Combined with the *enhanced Gaussian initialization*, which provably circumvents BPs via a depth-dependent variance and random local Clifford twirling during dataset construction. This study remains limited to classical simulators and benchmark circuits do not exceed $\sim 100$ qubits, however, in the future, we plan to scale to deeper circuits in real-world quantum sensors, ultimately enabling resource-efficient VQE deployments for chemically and physically relevant systems.

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
