# OpenReview forum: "TITAN: A Trajectory-Informed Technique for Adaptive Parameter Freezing in Large-Scale VQE"
_NeurIPS.cc/2025/Conference — NeurIPS 2025 poster_

### Official Review · Reviewer_cy7n · 2025-06-03

**Clarity:** 3
**Significance:** 3
**Originality:** 3
**Rating:** 4
**Confidence:** 5

**Summary:**

TITAN is the work proposed in this paper to reduce the number of measurements required by VQEs by freezing some of the parameters statically and a priori so that the parameter shift doesn't have to be applied for their gradient calculation, saving circuit runs, time, and resources. The work also supports the prior work in reducing barren plateaus by showing that parameter freezing can also help reduce them in the optimization landscape. TITAN trains a RestNet model with input features as circuit parameters and their freezing intensities and outputs which parameters should be frozen before the VQE is executed.

**Questions:**

How were 100-qubit circuits evaluated using simulators when it is impossible to simulate them on even the largest supercomputers? Were they purely Clifford?

Would a different type of TITAN model need to be trained for different types of ansatz? An alternative to HEA, for example?

What is the overall overhead of training and all the measurements that need to be done for training? Is the saving in inference worth it if each trained model only serves a limited class of VQE instances?

Did you observe any interesting trends in which parameters got frozen (e.g., beginning layers, end layers, etc.)?

Would TITAN need a large training dataset? Does a new model have to be trained for different ansatz sizes, number of qubits, number of layers, etc.? Do different VQE domains require different Titan models?

**Ethical Concerns:**

["NO or VERY MINOR ethics concerns only"]

**Final Justification:**

The rebuttal has provided additional results (the exact minimum energies, etc.), which have clarified many of my questions. Therefore, I have updated my score.

**Limitations:**

The paper does not have direct negative societal impacts. Some limitations of the work are discussed, such as the scaling issues; others are highlighted above in the form of questions and weaknesses.

**Paper Formatting Concerns:**

The paper has no major formatting concerns.

**Quality:**

2

**Strengths And Weaknesses:**

Strengths

Parameter freezing of VQE is a bit of an underexplored area, and the thought of freezing the parameters a priori is especially interesting.

The empirical analysis indicates that there is a large scope for parameter freezing without losing performance in terms of achieving the ground state energy of the VQE Hamiltonian.

Weaknesses:

We don’t know if the baseline technique reaches the minimum eigenvalue (the ground state energy in VQE) since only the difference between the baseline and TITAN is shown. Both techniques could be performing poorly and have a low differential. Please provide the results as raw numbers or provide the differential as a percentage of the raw minimum eigenvalue.

No error bars are provided for any values, even though they are all statistical quantities. For instance, the intensity of frozen parameters is a fractional quantity that varies from ansatz to ansatz, even if the two ansatzes have the same number of qubits (e.g., in Figure 2). So, some sort of variation metric or visualization should be provided, as these are not deterministic numbers. Also, Figure 2 just conveys that the number of frozen parameters increases linearly with the number of qubits and the number of layers. I'm not sure if this is a novel insight.

Shouldn’t the Hamiltonian itself be an input to the TITAN model training (currently, the input is only circuit features and freezing intensity of the parameters), because not just the circuit features but also the optimization loss affect the frozen parameters? So, the Hamiltonian itself should be included as an input? This might also help generalize the model. A similar case can be made for initial parameter values. They also affect the freezing intensity and should be included in the training.

The frozen parameters vary dynamically over time. I'm not sure if it is the best idea to freeze them statically a priori.

In most cases, the number of parameters frozen seems to be very low (Table 1), having a negligible impact on measured parameters.

---

> ### Author Rebuttal · Authors · 2025-07-28
>
> We sincerely thank Reviewer cy7n for the constructive feedback. We have carefully addressed the concerns raised under **Weaknesses** [**W1–W5**] and the specific **Questions** [**Q1-Q5**]. We hope the following responses effectively address the reviewer’s concerns and aid in the reassessment of our submission.
>
> **W1: Report the raw energy values or the difference as a percentage of the ground-state energy.**
>
> **Reply**: We have followed the reviewer's advice to exhibit how far TITAN and the baseline models deviate from the ground-state energy. For convenience, Table R9 summarizes the corresponding key findings. Please refer to Table R3 (reply to **W1** for Reviewer 8G53) for details.
>
> **Table R9**: Energy differences between Molecules with three initialization strategies and TITAN Freezing.
> |           | H₂ (4)   | HF (10)   | LiH (10) | BeH₂ (12) | H₂O (10)  | N₂ (12)   | CO (12)   |
> |---------|--------:|--------:|--------:|---------:|-------:|-------:|-------:|
> | Gaussian [1]| -0.896 $\pm$ 0.106| -97.601 $\pm$ 0.114 |  -7.882 $\pm$ 0.085 |  -14.876 $\pm$ 0.506 | -72.876 $\pm$ 2.065| -103.411 $\pm$ 2.639 | -107.669 $\pm$ 3.095|
> | Gaussian [1] + TITAN | -0.896 $\pm$ 0.106 |  -97.601 $\pm$ 0.114 | -7.864 $\pm$ 0.085| -14.868 $\pm$ 0.522 | -72.876  $\pm$ 2.065 | -103.418 $\pm$ 2.891| -107.684  $\pm$ 3.001 |
> | Exact| -1.175  |-100.340 | -8.023 |-15.918 | -76.428  | -109.542  | -113.326  |
> | TITAN Frozen Params |  1/3  |  1/3  |   10/24  |   21/92 |   4/54 |   47/117|   77/117  |
>
> The achieved results in Table R9 provide the following insights:
> 1) For most cases, the baseline model [1] and TITAN are very close to the exact results;
> 2) For the molecule CO, TITAN achieves an additional energy reduction of approximately 0.015 while halving the number of parameters.
> 3) The slight discrepancy between TITAN and the exact results stems from **the choice of ansatz**. For example, in the 30-qubit TFIM experiment, TITAN can freeze 30% of the parameters while still achieving convergence comparable to the exact result, as shown in Table R10.
>
> **Table R10**: Statistical performance of TITAN on diverse VQE benchmarks (Simplified version of Table R5)
> | Train | Test | Avg $\Delta E = E_{TITAN} - E_{Gaussian}$ ( $\Delta E / E_{Gaussian}$) | Avg Saved Paras |
> |------|------|-------|------|
> | HEA (Isotropic Hamiltonian) | HEA (Isotropic Hamiltonian)   | $ ‑0.0678 \pm 0.3851 $ (-0.48 \% $\pm$ 0.92\%)  |  $ 4.07 \pm  4.29 $ (\%) |
> |  HEA (Anisotropic Hamiltonian) | HEA (Anisotropic Hamiltonian)  |  $‑0.0078 \pm 0.1851 $  ( -0.80 \%  $\pm$ 1.28\%) |   $ 4.05 \pm  4.31$ (\%)|
>
> [1] Zhang et al. Escaping from the barren plateau via Gaussian initializations in deep variational quantum circuits. NeurIPS (2022).
>
> **W2: No error bars are provided in Figure 2.**
>
> **Reply**: We would like to remind the reviewer that Figure 2 is intended **solely to illustrate the 'parameter-freezing' phenomenon that motivates our study**. For ease of visualization in the 3D plot, uncertainty bands were omitted. To support the illustration with statistical rigor, we have included detailed results in the Appendix, based on ten independent runs with different random seeds.
>
> In addition, we would like to emphasize that all performance figures already include statistical variation in the submission, i.e., Figs. 5-7, as well as newly added results (Table R1-R11).
>
> Regarding the concern "*the number of frozen parameters increases ...,  if this is a novel insight*", we believe this is indeed a **novel** observation. To the best of our knowledge, the freezing phenomenon has not been explicitly characterized in prior work. Notably, *this novelty was also acknowledged by Reviewer 8G53*. Moreover, this observation serves as the foundation for TITAN, whose effectiveness has been systematically validated through extensive simulations.
>
> **W3: Shouldn’t the Hamiltonian itself be an input to the TITAN model training?**
>
> **Reply**: We would like to emphasize that whether the Hamiltonian itself should be encoded as input to TITAN depends on **the nature of the Hamiltonians under consideration**. In our submission, we focus on Heisenberg and TFIM models, which share a common Hamiltonian structure and therefore offer limited additional information to encode. As such, we do not include the Hamiltonian structure as input.
>
> However, we fully agree that for more complex lattice systems, e.g., triangular, hexagonal, or long-range interacting graphs, the Hamiltonian encodes rich structural information that may strongly influence parameter freezing behavior [1–2]. We have added a discussion of this direction to the updated version as a promising avenue for future work.
>
> [1] Yang et al. Scalable variational Monte Carlo with graph neural ansatz.
>
> [2] Roth et al. Group convolutional neural networks improve quantum state accuracy.
>
> **W4: Is it appropriate to freeze them statically at initialization?**
>
> **Reply**: We would like to emphasize that the choice between static and dynamic parameter freezing is **not unique**, as each strategy has its own merits and trade-offs. In particular, a dynamic freezing strategy may incur significant computational overhead, as updating the freeze mask at each iteration requires re-measuring gradients and re-synchronizing the optimizer. Accordingly, this cost scales linearly with both the number of parameters and the shot budget.
>
> We consider TITAN's static approach a practical first step, supported by strong empirical results (e.g., Table R5). That said, dynamic or hybrid strategies remain promising directions for future work, now noted in the revised manuscript.
>
> **W5: In most cases, the number of parameters frozen seems to be very low.**
>
> **Reply**: We acknowledge that our current presentation may not have sufficiently highlighted the statistical advantages of TITAN over baseline methods, which might have contributed to the reviewer's impression that the improvements are marginal. To address the reviewer's concern, in the following, we briefly introduce the main reasons.
>
> 1) Regarding the marginal improvement in Table 1. The number of frozen parameters is **highly dependent on $\tau$ and Hamiltonian**. With the results in Table R9, it is notable that for CO and N2, more than 50\% parameters are frozen.
>
> 2) As elucidated in Table R5, TITAN often attains an evident improvement for **In-Distribution tasks**. Even for **domain-generalization** tasks, TITAN can provide notable enhancement, enabling a broader applicability.
>
> 3) As shown in Table R6, the benefit of TITAN increases rapidly as problem instances grow larger.
>
> **Q1: How were 100-qubit circuits simulated—were they Clifford-only?**
>
> **Reply**: We would like to point out that efficient simulation of large-scale quantum circuits, those exceeding 100 qubits, is an active area of research. Recent advances in *tensor-network simulators* and *Pauli propagator techniques* [Sci. Adv.10,eadk4321(2024)] have enabled the simulation of specific classes of non-Clifford circuits. For instance, [1] demonstrates the simulation of quantum convolutional neural networks with up to **1,024 qubits**.
>
> [1] Bermejo et al. "Quantum convolutional neural networks are (effectively) classically simulable."
>
> The main challenge in evaluating TITAN beyond 30 qubits lies in the **demands of VQE optimization**. While large-qubit circuits can be simulated, existing simulators are not yet efficient enough to support full optimization workflows, i.e., collecting the rich gradient trajectories required for training. Any progress in this direction would broaden TITAN’s applicability.
>
> **Q2: Would a different type of TITAN model need to be trained for different types of ansatz? An alternative to HEA?**
>
> **Reply**: TITAN demonstrates strong domain generalization capability. As shown in Table R5 (response to **W1** for Reviewer 27vu) and Table R11, in **domain generalization tasks**, TITAN is trained on a family of Hamiltonians using HEA and then applied (without retraining) to structurally distinct ansätze such as SU2 and SEL by directly predicting their frozen parameters without increasing final energy. For an anisotropic Hamiltonian, which is more complex due to the coupling strengths differing between directions, TITAN consistently attains better energy estimations even when parameters are frozen, underscoring its domain generalization ability.
>
> **Table R11** (Simplified Table R2): Performance of $\Delta E$ on domain generalization tasks.
> | Testing Ansatz | N5 L5| N15 L5| N15 L15 |
> |:-----:|--------|--------|---------|
> | SEL | + 0.4415 $\pm$ 0.0541 |  -0.6715 $\pm$ 1.1730 | -0.5213 $\pm$  0.2285|
> | SU2  | + 0.1231 $\pm$ 0.3681 | -0.3110 $\pm$ 1.7138 | -0.3013 $\pm$ 0.0351 |
>
> **Q3: What is the overall overhead of training and all the measurements that need to be done for training?**
>
> **Reply**: We would like to accord Table R7 (**Q2** for Reviewer 27vu) to address the reviewer's concern. Once TITAN is optimized, no further training is required. As the number of test samples increases and the scale becomes larger, TITAN's benefits become more significant.
>
> **Q4: Any interesting trends?**
>
> **Reply**: We observe that the freezing phenomenon exhibits problem-dependent patterns. In Figure 4 (main text) and Figures 6, 7, 9, 12, 16, and 17 (Appendix), the distribution of frozen parameters varies across different ansätze and Hamiltonians. To capture these variations, we propose TITAN as a data-driven solution that adaptively selects freeze candidates, enabling it to generalize effectively across HEA, SU2, and SEL circuits (see the reply of **Q3** for Reviewer 27vu ).
>
> **Q5: Dataset & cross‑ansatz generalization**
>
> **Reply**:  Let us address the reviewer's concerns in sequence.
> 1) TITAN does not require a large dataset as referred to in the response to **W1** for Reviewer 27vu.
> 2) We further evaluated the HEA‑trained model outside the training distribution, as referred to in the response to **W2** for Reviewer dadQ.

---

> ### Author Response · Authors · 2025-08-04
> **Thank You for Your Positive Review Update**
>
> Dear Reviewer,
>
> Thank you very much for your positive update and for taking the time to review our additional results. We appreciate your constructive feedback throughout the process!
>
> Thank you!

---

### Official Review · Reviewer_27vu · 2025-06-27

**Clarity:** 3
**Significance:** 3
**Originality:** 3
**Rating:** 4
**Confidence:** 5

**Summary:**

This paper focuses on the expense related to sampling overhead in VQE approaches on quantum computers, especially as it pertains in scaling with respect to number of parameters as general backpropagation is known not to be possible due to consequences of the no-cloning theorem.  Hence, reducing the number of actively trained parameters has the potential to dramatically reduce overall training costs for a commonly used method, which is exploited here.  Previous strategies have focused on simple parameter elimination or layerwise freezing.  This work introduces a new method based on machine learning from data on other optimization runs in order to predict optimized, adaptive freezing schedules to reduce costs while maintaining speed and accuracy as much as possible.  Across a range of systems up to 30 qubits, it achieves faster convergence and 40-60% fewer circuit evaluations to reach or surpass current accuracy.

**Questions:**

1. Can you try to better quantify in a single plot the typical expected improvement (or penalty) for time and energy when using Titan?  Right now it seems like there is substantial chance the energy gets worse in 50% of the cases for a modest decrease in cost, but I can't quite tell because it's in so many different plots for so many different models.  A summary statstic showing a high likelihood of maintaining quality, while clearly stating averaged reduced resources would be helpful for improving the significance and quality scores.

2. Can you say more specifically what models and trajectories the Titan model itself was trained on, how much data was used, and what the training time was like?  Were the models exactly those tested in the paper, but with different parameters? Were they totally unrelated? If they were totally unrelated, or at much smaller sizes, this would strengthen the case this approach would generalize.

3. The plotted freezing intensity suggests a somewhat deterministic pattern being discovered.  How does Titan compare to a freezing strategy that is not uniformly random, but rather. freezes early layers with much higher probability than later layers? Does Titan substantially beat such a baseline? See Figure 4 for my intuition.

**Ethical Concerns:**

["NO or VERY MINOR ethics concerns only"]

**Final Justification:**

Initial concerns about if there was any generalization have been addressed, and it is more clear that the method offers broad improvements.  There is some room for additional quantification of end to end advancement, but broadly I would now lean towards accepting.

**Limitations:**

The authors could perhaps benefit from
1. Further discussing the costs of training their approach to get a holistic view of cost
2. Mention the additional problems in variational circuits related to local minima in the cost function

**Quality:**

3

**Strengths And Weaknesses:**

Strengths

The paper is clear in the setup of the problem and motivation.  The conceptual novelty of using an ML approach to help decide the freezing strategy and optimize for cost is interesting, and different than previous metalearning approaches for guessing the parameters outright.

Weaknesses

The major weakness of the paper is that for the added implementation complexity, the improvements are not as striking as one would hope.  In many of the presented results, Titan does a little bit worse or about the same as much simpler approaches, and only decreases the cost by a small constant factor.  It certainly appears to help in some cases, but it looks like it's not very obvious at the outset when it will, so for near experiments, one is likely to have to run the baseline anyways.  The results are presented in a way that make it a bit hard to see the potential advantages, and there is not much discussion of the size of training set or time required to reach the improvements reported.

---

> ### Author Rebuttal · Authors · 2025-07-28
>
> We sincerely thank Reviewer 27vu for handling our submission and providing constructive feedback. We have carefully addressed the concerns outlined under **Weaknesses** [**W1**], specific **Questions** [**Q1-Q3**], and **Limitations** [**L1-L2**]. We hope the responses below clarify our contributions and support a more informed reassessment of our submission.
>
> **W1: (1) The added implementation complexity of TITAN does not seem to yield significant performance gains over simpler baselines. (2) TITAN’s benefits appear inconsistent across settings, making it unclear when it will be effective. (3) The results lack clarity in highlighting TITAN’s advantages, and there is insufficient discussion of training set size and runtime overhead.**
>
> **Reply**: To ensure clarity, we address the reviewers’ three concerns in sequence below.
>
> (1) *TITAN does not seem to yield significant performance gains over simpler baselines*. We acknowledge that our current presentation may not have sufficiently highlighted the **statistical advantages of TITAN** over baseline methods, which might have contributed to the reviewer's impression that *the improvements are marginal*. To address this, we have reorganized the simulation results in the main text and now present the supporting statistics in Table R5.
>
> **Table R5**: A summary of the statistical performance of TITAN. **DG** and **ID** refer to *Domain Generalization* and *In-Distribution*, respectively. Isotropic means equal couplings; anisotropic means direction-dependent couplings Hamiltonian. Energy difference $\Delta E$ is defined in Table R1.
> | Train | Test | Avg $\Delta E = E_{TITAN}- E_{Gaussian}$ | Avg Saved Paras | Task type|
> |---------|-----|-------|----|:----:|
> | HEA (Isotropic Hamiltonian) | HEA (Isotropic)   | $ ‑0.0678 \pm 0.3851$  |  $ 4.07 \pm  4.29 $ (\%) | ID |
> | HEA (Isotropic) |  SU2 (Isotropic)  |  $‑0.1982 \pm 0.3880 $ | $ 2.96 \pm  6.33$ (\%)| DG |
> |  HEA (Isotropic) |  SEL (Isotropic) | $‑0.0933 \pm 0.1911 $ | $ 3.89 \pm  4.12$ (\%) |  DG |
> |  HEA (Anisotropic) | HEA (Anisotropic)  |  $‑0.0078 \pm 0.1851 $ | $ 4.05 \pm  4.31$ (\%)| DG |
>
> **Sub-Table R5**:  VQE benchmarks for molecules (i.e., H2 , HF, LiH , BeH2, H2O, N2, CO).
>
> | Train | Test  | Avg $\Delta E = E_{TITAN}- E_{Gaussian}$ | Avg Saved Paras | Task type|
> |-------|------|--------|-------|:----:|
> | Molecules | Molecules | $ + 0.00057  \pm  0.000094 $  |   $ 34.94  \pm  3.226 $ (\%)| ID |
>
> We would like to clarify that the tasks in Table R5 fall into two broad categories: **in-distribution** and **domain generalization** tasks. The in-distribution tasks involve predicting frozen parameters for unseen Hamiltonians within the same family, using a fixed ansatz. In contrast, the domain generalization tasks evaluate TITAN’s ability to transfer across ansätze: the model is trained on a family of Hamiltonians using the HEA, and then applied to structurally distinct ansätze (i.e., SU2 and SEL) without retraining, by directly predicting their frozen parameters. While domain generalization is inherently more challenging, it demonstrates TITAN’s broad applicability better, enabling a single model to generalize across a wide range of VQE tasks.
>
> The achieved results in Table R5 indicate that TITAN achieves *substantial improvements over baseline methods on in-distribution tasks*. Specifically,  more than 4% parameters are saved while maintaining more accurate ground energy, and saved parameters are up to an average of 36% for molecule tasks in Sub-Table 5. Even for domain generalization tasks, it provides notable improvements over baseline models that lower the ground energy with an average of 3.5% parameters saved.
>
> (2) *It is unclear when TITAN will be effective*.  The results in Table R5 indicate that an optimized TITAN model can benefit a wide range of VQEs across diverse ansätze and Hamiltonians. This broad applicability allows TITAN to reduce measurement overhead, with its advantages compounding as *the number of inference-time Hamiltonians increases*.
>
> A concrete example is advancing **quantum phase classification** [1]. For an $N$-qubit TFIM, we need to sweep $n_J$ couplings and $n_h$ field parameters to determine the critical point. As a result, the total number of TFIMs to be explored is  $n_{grid} = n_J n_h$. For each TFIM, the number of Pauli terms is  $K = 2N-1$. Assume the employed ansatz contains $P$ trainable parameters and the parameter-shift rule is used, which requires two circuit evaluations per parameter for each Pauli term. For each evaluation, $10^3$ shots are required. Taken together, the measurement cost for each TFIM is $ M_0 = 2KP_\theta = 2\times 10^3 (2N-1) P$ $. For $n_{grid}$ TFIMs, the vanilla VQE takes $ M_{tot} =  n_{grid} M_0 = 2\times 10^3 (2N-1) n_J n_h P$ for a single iteration. We show the quantitative analysis in Table R6 below, which exhibits *the benefit of TITAN compounds rapidly as problem instances grow larger*.
>
> **Table R6**:  Baseline measurement cost for HEA ansatz in 1D-TFIM phase classification when layer number is $l = 6$ with 5% parameters freezing for TITAN. Here we only consider a single iteration.
> | Qubits N | $n_J$  | $n_h$ | $P $ | $ M_{tot} $ | $P^{TITAN} = 0.95 P$ | Reduced $ M_\mathrm{tot} $ |
> |:------:|:------:|:-----:|:------:|:-----:|:-----:|:-----:|
> | 10 | 50 | 50 | 120 | $1.14×10^{10}$  | 114 | $5.7×10^8$ |
> | 100 | 50 | 50 | 1200 | $1.19×10^{12}$  | 1140 | $5.95×10^{10}$  |
> | 1000 | 50 | 50  | 12000 | $1.20×10^{14}$ | 11400 |  $6×10^{12} $  |
>
> [1] Herrmann et al. Nat Commun 13, 4144 (2022).
>
> (3) *What are the advantages of TITAN? How about its training set size and runtime overhead?*
>
> The required training dataset size and runtime overhead of TITAN are summarized in Table R7. Following explanations in **W1 (2)**, the computation cost in the training is ignorable, particularly when amortized over a growing number of inference-time Hamiltonians. Moreover, TITAN’s advantages compound as the number of such inference tasks increases.
>
> **Table R7**: Training details of TITAN.
> torch: 2.5.1+cu121| NVIDIA GeForce RTX 3060 12 GB
> | Dataset | Details| Samples per Case |Dataset Size | Training Time per Epoch |
> |---------------|--------|:-------:|-----|--------|
> | Isotropic Hamiltonian |  N(5-15), L(5-15) | 20 | 79.2 MB | 24.056 $\pm$ 2.671 s |
> | Anisotropic Hamiltonian |  N(8), L(5-7), (a,b,c) in [-5,5]| 20 | 527 MB | 647.564 $\pm$ 11.389 s|
> | Molecule |  H2 , HF, LiH , BeH2,  H2O, N2, CO |20 | 2.38 MB | 1.345 $\pm$ 0.284 s |
>
> **Q1: Provide a single plot summarizing the typical expected improvement.**
>
> **Reply**: Due to figure upload restrictions, we present the results in Table R5. As noted in our response to **W1 (1)**, TITAN achieves an average parameter freeze of 3–5% on most **in-distribution tasks** while maintaining the accuracy of the minimum energy solution. Notably, for larger molecules or large-scale VQE instances, the freeze ratio can reach up to 60%. For **domain generalization** tasks, TITAN can still provide meaningful improvements.
>
> **Q2: (1) Specify training details used for TITAN. (2) Were the training models the same as those tested? If they were unrelated or smaller in scale, it would further support the generalizability of the approach.**
>
> **Reply**:  Let us address the reviewer's two concerns in sequence.
>
> **Concern (1)**. As suggested, we have included additional implementation details of TITAN and have provided the code and dataset in the supplementary materials. For illustration, Table R7 outlines the data provenance, network hyperparameters, and training time used.
>
> **Concern (2)**. The reviewer's understanding is correct. As shown in **Tables R2 and R5**, **TITAN exhibits strong domain generalization across ansätze**. Specifically, the model is trained on a family of Hamiltonians using HEA, and is then successfully applied (without retraining) to predict frozen parameters for structurally distinct ansätze such as SU2 and SEL, achieving competitive performance.
>
> **Q3: How does TITAN compare to a heuristic that freezes early layers with higher probability?**
>
> **Reply**:  We would like to clarify that the distribution of frozen parameters is **problem-dependent**. While Figure 4 illustrates a tendency for frozen parameters to concentrate in the shallow layers when using HEA on isotropic Hamiltonians, this pattern is **not universal**. As shown in Figures 6, 7, 16, and 17 of the Appendix, the distribution varies substantially across different ansätze and problem Hamiltonians.
>
> To better address the reviewer's concern, we compare TITAN against the Early-Layer Freezing (ELF) heuristic, as suggested. The achieved results, as summarized in Table R8, indicate that the TITAN freezing strategy shows faster and more thorough gradient decay at all qubit scales (N=5–30).  While ELF beats plain Random, it still trails TITAN, and the gap widens as circuit size increases, around 4 times higher than TITAN in gradient norm and 3.1% higher final energy.
>
> **Table R8**: Comparison of gradient norm and final energy between TITAN and ELF for 30-qubit TFIMs.
>  | Iteration |  0  | 40 |  80 | Final Energy |
> |-----------|----|---|----|----|
> | Baseline | 10.024 $\pm$ 1.519| 3.924 $\pm$ 1.104 |   0.182 $\pm$ 0.095| -89.364 $\pm$ 1.500  |
> | Random |9.107 $\pm$ 1.262| 9.718 $\pm$ 1.054 |   3.055 $\pm$ 1.390|  -51.169 $\pm$ 2.267  |
> | TITAN | 8.766 $\pm$ 1.259 | 6.624 $\pm$ 1.073| 0.383 $\pm$ 0.200|  -86.156 $\pm$ 2.022  |
> | ELF | 11.677 $\pm$ 1.853 | 5.465 $\pm$ 2.791 | 1.310 $\pm$ 1.283 |   -83.494 $\pm$ 3.758 |
>
> **L1 & L2: Discuss training cost and problems related to optimizing VQAs.**
>
> **Reply**: We have added a discussion on the training cost of TITAN; please refer to our response to **Q1** for details. In addition, we have included more references in the updated manuscript related to VQA optimization, covering topics such as barren plateaus, measurement reduction, and the local minima landscape.

---

> > ### Author Response · Authors · 2025-08-03
> > **Kind Request for Further Feedback on Our Rebuttal**
> >
> > Dear Reviewer,
> >
> > I hope this message finds you well. As the rebuttal deadline is approaching, we would greatly appreciate your valuable feedback at your earliest convenience.
> >
> > I would like to kindly ask whether our latest responses adequately addressed your concerns and clarified the points raised in your previous question, **including (i)** the unified statistical summary in **Table R5** that makes TITAN’s performance gains explicit, **(ii)** the new comparison against the early-layer-freezing heuristic in **Table R8**.
> >
> > Your detailed and constructive feedback has been invaluable in shaping our work, and we deeply value your insights.
> >
> > Our primary goal is to ensure that the paper meets the rigorous standards of NeurIPS. If there are any remaining concerns or areas where further clarification might help, we would be more than happy to address them in the spirit of collaboration and continuous improvement.
> >
> > Thank you for your time and guidance!

---

> > > ### Comment · Reviewer_27vu · 2025-08-04
> > >
> > > Thank you for your detailed response.  The additional clarifications on improvement and more information on how it is generalizing to different ansatz improves my belief in the significance of the work.  I have will my scores positively.

---

> ### Author Response · Authors · 2025-08-04
> **Thank You for Your Positive Review Update**
>
> Dear Reviewer,
>
> Thank you very much for your feedback and for updating your scores positively! Your thoughtful comments are important in refining our manuscript, and we greatly appreciate the time and care you have devoted to the review process.
>
> Thank you again for your constructive support!

---

### Official Review · Reviewer_8G53 · 2025-07-03

**Clarity:** 4
**Significance:** 3
**Originality:** 3
**Rating:** 5
**Confidence:** 2

**Summary:**

To tackle challenges faced by Variational quantum Eigensolver (VQE), this paper introduces a deep learning framework called TITAN. The proposed method detects and freezes inactive parameters of a given ansatz at initialization for a targeted class of Hamiltonians.

**Questions:**

See Strengths And Weaknesses

**Ethical Concerns:**

["NO or VERY MINOR ethics concerns only"]

**Limitations:**

Yes

**Quality:**

3

**Strengths And Weaknesses:**

S1. The proposed method TITAN significantly speeds up the convergence, uses fewer circuit evaluations, while maintaining the accuracy. The methodology and experimental results demonstrate a non-trivial contribution.

S2. This paper is well-written and logically structured, making it accessible and easy to read even for readers who are not experts in the field.

S3.  The appendix and codebase are clear and well-structured, providing rich and important information.

W1. Can this method efficiently achieve accurate results for larger molecules?

W2. The frozen parameter is innovative, but the model architecture shows limited novelty.

---

> ### Author Rebuttal · Authors · 2025-07-28
>
> We sincerely appreciate Reviewer 8G53 for the thoughtful and constructive feedback. We have carefully addressed the points raised under **Weaknesses** [**W1-W3**], and the specific **Questions** [**Q1**]. We hope the following responses help clarify our contributions and assist in the reviewer's reassessment of our submission.
>
> **W1: Can this method efficiently achieve accurate results for larger molecules?**
>
> **Reply**: We agree with the reviewer's viewpoint to understand the scalability of TITAN. To address this issue, we have conducted extensive experiments to study the **scalability of Titan** in the original submission. In particular, the results in Figure 7 (Page 9) showed that TITAN could be transferred to models other than those trained up to **30 qubits** and still maintain the same performance level as the baseline.
>
>
> **Table R3**: Energy differences with the ground truth across different molecules over three initialization strategies and TITAN Freezing.
> |             | H₂ (4)   | HF (10)   | LiH (10) | BeH₂ (12) | H₂O (10)  | N₂ (12)   | CO (12)   |
> |-------------|:---------:|:---------:|:---------:|:----------:|:----------:|:----------:|:----------:|
> | Gaussian [1]| -0.896 $\pm$ 0.106| -97.601 $\pm$ 0.114 |  -7.882 $\pm$ 0.085 |  -14.876 $\pm$ 0.506 | -72.876 $\pm$ 2.065| -103.411 $\pm$ 2.639 | -107.669 $\pm$ 3.095|
> | Zero  | -0.896 $\pm$ 0.106 | -97.601 $\pm$ 0.114 |  -7.882 $\pm$ 0.085 |  -14.876 $\pm$ 0.506 | -72.876 $\pm$ 2.065| -103.411 $\pm$ 2.639 | -107.669 $\pm$ 3.095|
> | Uniform| -0.639 $\pm$ 0.088| -97.389 $\pm$ 0.201 |  -7.401 $\pm$ 0.107 |  -13.743  $\pm$ 0.932 | -72.281 $\pm$ 2.107 | -102.616 $\pm$ 4.898 | -107.217 $\pm$ 3.209|
> | Gaussian [1] + TITAN | -0.896 $\pm$ 0.106 |  -97.601 $\pm$ 0.114 | -7.864 $\pm$ 0.085| -14.868 $\pm$ 0.522 | -72.876  $\pm$ 2.065 | -103.418 $\pm$ 2.891| -107.684  $\pm$ 3.001 |
> | Zero + TITAN | -0.896 $\pm$ 0.106 | -97.601 $\pm$ 0.114 | -7.899 $\pm$ 0.077 | -14.884 $\pm$ 0.730 | -72.876 $\pm$ 2.065 | -103.404 $\pm$ 2.908| -107.654 $\pm$ 3.031|
> | Uniform + TITAN | -0.142 $\pm$ 0.577 | -97.537 $\pm$ 0.197 | -7.256 $\pm$ 0.131 | -13.979 $\pm$ 0.984 | -72.359 $\pm$ 2.299 | -100.104 $\pm$ 5.424 | -108.946 $\pm$ 4.025|
> | Exact| -1.175 [1] |-100.340 [2]| -8.023 [2]|-15.918 [3] | -76.428 [4] | -109.542 [5] | -113.326 [5] |
> | TITAN Frozen Parameters |  1/3  |  1/3  |   10/24  |   21/92 |   4/54 |   47/117|   77/117  |
>
> To further address this issue, we explore the capability of Titan in seven quantum chemistry molecules under frozen threshold $\tau = 80$ in **Table R3**.  For CO, which involves 12 qubits, TITAN succeeds in freezing more than $60\%$ of those parameters while maintaining or improving the energy estimate.
>
> *The challenge towards larger molecules is the lack of **efficient simulators** for large-qubit circuits with optimization*. A detailed explanation in Reply to **Q1** for Reviewer cy7n. In words, with the development of tensor network [7] and CUDA-Q [8], computing resource limitations will be better resolved, and the applicability of TITAN can be further broadened.
>
> We appended the discussion to the revised manuscript.
>
> [1] Sims J S, Hagstrom S A. High precision variational calculations for the Born-Oppenheimer energies of the ground state of the hydrogen molecule[J]. The Journal of chemical physics, 2006, 124(9).
>
> [2] Lathiotakis N N et al. A functional of the one-body-reduced density matrix derived from the homogeneous electron gas: Performance for finite systems[J]. The Journal of chemical physics, 2009, 130(6).
>
> [3] Goldberg M C, Riter Jr J R. Confirmation of the predicted Hartree-Fock limit in BeH2[J]. The Journal of Physical Chemistry, 1967, 71(9): 3111-3112.
>
> [4] Gurtubay I G, Needs R J. Dissociation energy of the water dimer from quantum Monte Carlo calculations[J]. The Journal of chemical physics, 2007, 127(12).
>
> [5] Chachiyo T, Chachiyo H. Understanding electron correlation energy through density functional theory[J]. Computational and Theoretical Chemistry, 2020, 1172: 112669.
>
> [6] Zhang K et al. Escaping from the barren plateau via gaussian initializations in deep variational quantum circuits[J]. Advances in Neural Information Processing Systems, 2022, 35: 18612-18627.
>
> [7] Kulshrestha et al. Qadaprune: Adaptive parameter pruning for training variational quantum circuits.
>
> [8] Schieffer et al. Harnessing CUDA-Q’s MPS for Tensor Network Simulations of Large-Scale Quantum Circuits.
>
> **W2: The frozen parameter is innovative, but the model architecture shows limited novelty.**
>
> **Reply**: We appreciate the reviewer for the positive affirmation about TITAN. In fact, we have systematically explored the effect of different architectures. The main conclusion is that: **Any reasonably expressive backbone** can be plugged into the pipeline and will learn the same gradient‑suppression pattern, confirming that the framework is **genuinely model‑free**. In order to help readers better understand this work, we weakened the discussion about neural architecture.
>
> **Table R4**: $\Delta E$ (Frozen Parameters) for a 1‑D nearest‑neighbor Heisenberg chain using the HEA ansatz, comparing different TITAN backbone architectures under frozen‑parameter evaluation.
> | Qubits | Architectures (L: 8)  | Frozen threshold $\tau$ = 90 | $\tau$ = 80 | $\tau$ = 70 | $\tau$ = 60 |  $\tau$ = 50 |
> |:-------:|:-------------------|:---:|:---:|:---:|:---:|:---:|
> | **10** | ResNet-18 |-0.13 $\pm$ 0.02 (5/160) | -0.08 $\pm$ 0.02 (11/160) | -0.03 $\pm$ 0.03 (30/160) | +0.49 $\pm$ 0.15 (60/160) | +0.69 $\pm$ 0.35 (81/160) |
> |        | Transformer  [1] | -0.15 $\pm$ 0.00 (5/160) | -0.19 $\pm$ 0.01 (12/160) | -0.07 $\pm$ 0.02 (28/160) | +0.35 $\pm$ 0.19 (57/160) | +0.77 $\pm$ 0.28 (86/160) |
> |        |EfficientNetV2‑S [2]| -0.03  $\pm$ 0.00 (4/160) | -0.04 $\pm$ 0.02 (11/160) | -0.12 $\pm$ 0.(23/160) | +0.22 $\pm$ 0.07 (36/160)| + 0.43 $\pm$ 0.24 (78/160) |
> |        | U-Net [3]  | -0.24  $\pm$ 0.03 (7/160) | -0.03 $\pm$ 0.03 (8/160) | +0.18 $\pm$ 0.11 (27/160) | +0.55 $\pm$ 0.21 (44/160) | +0.62 $\pm$ 0.65 (85/160)|
> | **11** | ResNet-18   | -0.03 $\pm$ 0.03 (4/176) | +0.04 $\pm$ 0.01 (2/176) | +0.45 $\pm$ 0.05 (57/176) | +0.69 $\pm$ 0.27 (95/176) | +2.14 $\pm$ 0.23 (122/176) |
> |        | Transformer | -0.16 $\pm$ 0.03 (6/176) | -0.06 $\pm$ 0.03 (6/176) | -0.07 $\pm$ 0.41 (48/176) | +0.07 $\pm$ 0.51 (69/176) | +1.56 $\pm$ 0.57 (113/176) |
> |        | EfficientNetV2‑S | -0.05 $\pm$ 0.00 (6/176)| -0.07 $\pm$ 0.03 (6/176)| +0.33 $\pm$ 0.08 (51/176) | +0.55 $\pm$ 0.21 (69/176) | +1.71 $\pm$ 0.44 (109/176) |
> |        | U-Net | -0.19 $\pm$ 0.08 (4/176) | +0.05 $\pm$ 0.00 (8/176) | +0.32 $\pm$ 0.11 (48/176) | +0.37 $\pm$ 0.49 (71/176) | +1.55 $\pm$ 0.61 (116/176) |
> | **12** | ResNet-18   |-0.02 $\pm$ 0.01 (9/192) | +0.06 $\pm$ 0.01 (21/192) | -0.01 $\pm$ 0.01 (42/192) | +0.04 $\pm$ 0.14 (65/192) | +1.49 $\pm$ 0.57 (101/192) |
> |        | Transformer| +0.01 $\pm$ 0.01 (9/192) | +0.03 $\pm$ 0.03 (21/192)| +0.12 $\pm$ 0.08 (41/192) | +0.35 $\pm$ 0.19 (63/192) | +1.71 $\pm$ 0.34 (108/192) |
> |        | EfficientNetV2‑S   | -0.06 $\pm$ 0.04 (8/192) | -0.01 $\pm$ 0.07 (19/192) | +0.15 $\pm$ 0.13 (42/192) | +0.28 $\pm$ 0.11 (68/192) | +1.34 $\pm$ 0.35 (96/192) |
> |        | U-Net   | -0.11 $\pm$ 0.09 (9/192) | -0.05 $\pm$ 0.00 (21/192) | +0.00 $\pm$ 0.01 (43/192) | +0.35 $\pm$ 0.15 (66/192) | +0.97 $\pm$ 0.71 (108/192) |
> | **15** | ResNet-18   | +0.43 $\pm$ 0.11 (1/240) | -0.03 $\pm$ 0.01 (8/240) | +0.58 $\pm$ 0.18 (27/240) | +0.86 $\pm$ 0.19 (72/240) | +1.50 $\pm$ 0.21 (128/240) |
> |        | Transformer   | +0.16 $\pm$ 0.19 (1/240) | -0.13 $\pm$ 0.06 (8/240) | -0.41 $\pm$ 0.27 (18/240)  | +0.39 $\pm$ 0.55 (61/240) | +1.45 $\pm$ 0.19 (125/240) |
> |        | EfficientNetV2‑S   | -0.03 $\pm$ 0.00 (1/240) | +0.06 $\pm$ 0.01 (7/240) | +0.33 $\pm$ 0.29 (25/240) | +0.65 $\pm$ 0.33 (69/240) | + 1.79 $\pm$ 0.19 (135/240) |
> |        | U-Net   | +0.23 $\pm$ 0.01 (1/240) | +0.08 $\pm$ 0.01 (8/240) | +0.42 $\pm$ 0.23 (29/240) | +0.61 $\pm$ 0.37 (69/240) | + 1.52 $\pm$ 0.17 (128/240) |
>
> To better address the reviewer's concerns, we retrained all substitutes—Transformer [1], EfficientNetV2‑S [2], and U‑Net [3] using the Adam optimizer with a fixed learning rate of $1\times10^{-3}$ for $1,000$ epochs, keeping batch size, loss function, and data augmentations identical to the default convolutional backbone (ResNet-18 in Table R4) in **Table R4**, where ResNet-18 based TITAN (model size 46 MB), Transformer based (345 MB), EfficientNetV2‑S based (90 MB), and U‑Net based (124 MB).
>
> Table R4 shows that every backbone yields energy differences $\Delta E$ that are **comparable** to the baseline TITAN model (ResNet-18), while freezing a similar fraction of parameters. The same trend holds across all qubit counts: swapping the backbone changes the numeric value by at most $2\times10^{-1}$, yet the qualitative behavior—larger $\tau$, more frozen parameters, and lower measurement cost with minimal loss in accuracy remains intact.
>
> We have added the new ablation study and accompanying discussion (Section 4) to highlight this architectural agnosticism and to clarify that the contribution of the paper is orthogonal to the choice of neural backbone in the revised manuscript.
>
> [1] Vaswani A, Shazeer N, Parmar N, et al. Attention is all you need[J]. 2017, 30.
>
> [2] Tan M, Le Q. Efficientnetv2: Smaller models and faster training[C]. PMLR, 2021: 10096-10106.
>
> [3] Ronneberger O, Fischer P, Brox T. U-net: Convolutional networks for biomedical image segmentation[C], 2015: 234-241.

---

> > ### Comment · Reviewer_8G53 · 2025-08-05
> >
> > Thank you very much for answering my questions with additional experimental results about the model architecture and the application to larger molecules! It's a solid work. I will keep my positive score.

---

> ### Author Response · Authors · 2025-08-05
> **Thank you for your positive feedback**
>
> Dear Reviewer,
>
> Thank you very much for your encouraging feedback and for maintaining your positive score! We appreciate the time you took to review our additional experiments. We are grateful for your constructive engagement throughout the review process!

---

### Official Review · Reviewer_dadQ · 2025-07-03

**Clarity:** 3
**Significance:** 2
**Originality:** 3
**Rating:** 4
**Confidence:** 3

**Summary:**

The paper first observes a phenomenon that in variational quantum circuits many parameters do not have drastic dynamics and hence can be frozen once optimized. Then the authors propose a learning-based framework to predict which parameters can be frozen to what values as initialization of variational quantum circuits. This prediction saves many resources in the training of variational quantum circuits and leads to faster convergence, which is demonstrated in experiments.

**Questions:**

For a frozen parameter, the reason that it does not move significantly is that the gradients tend to be zero for them. As observed in the paper, this phenomenon happens more in the shallower layers. I wonder if there is any connection between this phenomenon and the gradient vanishing problem in classical NN training.

**Ethical Concerns:**

["NO or VERY MINOR ethics concerns only"]

**Limitations:**

There are limited discussions on the limitation of the proposed method in the paper.

**Paper Formatting Concerns:**

No.

**Quality:**

3

**Strengths And Weaknesses:**

Pros
1. The observation of the phenomenon of frozen parameters is interesting and insightful.
2. With the observation, it is also interesting to propose a learning-based method to predict frozen parameters from the description of the variational task.
3. The experiments conducted are comprehensive and illustrative.

Cons
1. The data presented in Table 1 shows only a small portion of the parameters are frozen, possibly because of the high threshold. I would like to see how these thresholds affect the performance of the proposed method.
2. The color in Figure 6 is fainted for Random and TITAN, especially when printed. I suggest the authors make the transition smoother.
3. The details on how TITAN complements Zero initialization and others are missing. I cannot fully understand what Figure 5 means, since I do not know the experiment setting.

---

> ### Author Rebuttal · Authors · 2025-07-25
>
> We sincerely appreciate Reviewer dadQ for the thoughtful and constructive feedback. We have carefully addressed the points raised under **Weaknesses** [**W1-W3**], and the specific **Questions** [**Q1**]. We hope the following responses help clarify our contributions and assist in the reviewer's reassessment of our submission.
>
> **W1: How does the threshold $\tau$ affect the performance of TITAN?**
>
> **Reply**: Thanks for the comments. We fully agree with the reviewer's perspective that exploring the varied threshold settings can offer deeper insights into TITAN’s effectiveness. For this reason, in the *original manuscript*, we reported *how the varied thresholds (i.e., $\tau \in$ {50, 60, 70, 80, 90}) affect the performance of TITAN*, as elaborated in Tables 3-5 (pages 19-23, Appendix). The key observation is that as the threshold $\tau$ decreases, a larger proportion of parameters are frozen; however, this comes at the cost of reduced energy estimation accuracy.
>
> To better address the reviewer’s concern, we conducted additional simulations to support Table 1 in our manuscript. These experiments follow the same settings as in Table 1, with the only difference being the variation of the threshold values.
>
> The achieved results are shown in Table R1. In particular,  as $\tau$ is reduced from 90 to 50, the number of frozen parameters rises from at most $5$–$11$ per layer to more than half of the total, and higher ground state energy solution. These observations confirm that the proposed method is sensitive to $\tau$ in a predictable, monotonic fashion: lowering the threshold freezes more parameters and yields increasingly positive (i.e., worse) $\Delta E$, while excessively low thresholds offer limited additional improvement in single testing.
>
> **Table R1** (Supplementary Table 1) The performance of TITAN with the varied threshold $\tau$. The explored Hamiltonians are a family of 1‑D nearest‑neighbor Heisenberg models. The performance is evaluated by the energy difference, i.e., $\Delta E= E_{TITAN} - E_{Gaussian}$, where each $E$ denotes the estimated energy returned by TITAN-frozen Gaussian initialization and with standard Gaussian initialization, respectively.
>
> | Qubits | Metric (L: 8)  | $\tau$=90 | $\tau$=80 | $\tau$=70 | $\tau$=60 | $\tau$=50 |
> |:-------:|:----:|:---:|:---:|:---:|:---:|:---:|
> | **10** | $\Delta E$ (Frozen Params) |-0.13 $\pm$ 0.02 (5/160) | -0.08 $\pm$ 0.02 (11/160) | -0.03 $\pm$ 0.03 (30/160) | +0.49 $\pm$ 0.15 (60/160) | +0.69 $\pm$ 0.35 (81/160) |
> | **11** | $\Delta E$ (Frozen Params) | -0.03 $\pm$ 0.03 (4/176) | +0.04 $\pm$ 0.01 (2/176) | +0.45 $\pm$ 0.05 (57/176) | +0.69 $\pm$ 0.27 (95/176) | +2.14 $\pm$ 0.23 (122/176) |
> | **12** | $\Delta E$ (Frozen Params) | -0.02 $\pm$ 0.01 (9/192) | -0.03 $\pm$ 0.03 (21/192) | -0.01 $\pm$ 0.01 (42/192) | +0.04 $\pm$ 0.14 (65/192) | +1.49 $\pm$ 0.57 (101/192) |
> | **13** | $\Delta E$ (Frozen Params) | -0.04 $\pm$ 0.14 (2/208) | -0.07 $\pm$ 0.00 (8/208) | +0.16 $\pm$ 0.03 (19/208) | +0.31 $\pm$ 0.14 (43/208) | +0.76 $\pm$ 0.41 (78/208) |
> | **14** | $\Delta E$ (Frozen Params) | -0.08 $\pm$ 0.03 (5/224) | -0.08 $\pm$ 0.01 (6/224) | +0.15 $\pm$ 0.05 (20/224) | +0.43 $\pm$ 0.21 (45/224) | +0.69 $\pm$ 0.33 (96/224) |
> | **15** | $\Delta E$ (Frozen Params) | + 0.02 $\pm$ 0.24 (1/240) | -0.03 $\pm$ 0.01 (8/240) | +0.58 $\pm$ 0.18 (27/240) | +0.86 $\pm$ 0.19 (72/240) | +1.50 $\pm$ 0.21 (128/240) |
>
> In the updated version, we have incorporated the above clarification into Section 4 (“Sensitivity to $\tau$”) and reproduced Table 1 to make the relationship between threshold and performance explicit.
>
>
> **W2: The color in Figure 6 is faint for Random and TITAN.**
>
> **Reply**:  We have followed the reviewer's advice to redraw Figure 6 using a higher‑contrast and colour‑blind‑safe palette,  and increase both line weight and marker size to ensure clear differentiation in the revision.
>
> **W3: How does TITAN complement the zero initialization, and what does Figure 5 mean?**
>
> **Reply**: To fully address the reviewer's concerns, in this reply, we first show how Titan complements with other initialization methods, followed by elucidating the settings adopted in Figure 5 (page 8).
>
> $\bullet$ *Complementary with other initialization methods.* We would like to clarify that TITAN complements Zero and other initialization strategies in a manner analogous to how Gaussian and Uniform initialization methods operate, as demonstrated in Figure 5 (Page 8). Specifically, TITAN predicts in advance which parameters of a given ansatz are likely to be frozen, before any initialization. After this step, any standard parameter initialization strategy (e.g., Zero, Gaussian, Uniform) can be applied to the remaining trainable parameters.  We also showed how TITAN complements the Layer-wise Greedy Gradient Descent (LGGD) [1] method in Appendix Page 9. All explanations are appended to the revised manuscript.
>
> [1] Grimsley, Harper R., et al. "An adaptive variational algorithm for exact molecular simulations on a quantum computer." Nature Communications 10.1 (2019): 3007.
>
> $\bullet$ *Detailed settings in Figure 5.*  We would like to first remind the reviewer that the purpose of Figure 5 is to evaluate the **domain generalization ability of TITAN**. Specifically, TITAN is trained on a family of isotropic Hamiltonians using the hardware-efficient ansatz (HEA), and then tested on two **structurally distinct ansätze**, SU2 and SEL, by predicting their frozen parameters. This setup allows us to assess whether *a single TITAN model, trained on one ansatz, can generalize to others*, thereby demonstrating its broader applicability across different quantum circuit architectures.
>
> The experiment setup adopted in Figure 5 is as follows. For each ansatz (SU2 and SEL), we evaluate three commonly used initialization strategies, i.e., Gaussian, zero, and uniform, and examine how TITAN refines their corresponding freezing parameters. A purely random freezing mask is also included as a baseline for comparison. Five experimental settings are considered:  $(N,L)=(5,5),(5,15),(10,10),(15,5),(15,15)$, where $N$ and $L$ refer to the qubit count and the layer number, respectively.
>
> To make the outcome transparent, we have extracted the raw data from Figure 5 and **present it as the Table R2**. An observation is that in almost every setting, TITAN yields lower ground energies than the random freezing, confirming its effectiveness. Moreover, the low estimation error of TITAN on both SEL and SU2 validates its domain generalization capability, which is crucial for enabling robust and transferable performance across diverse ansatz architectures.
>
> **Table R2**: Evaluating TITAN’s domain generalization across distinct ansätze (restatement of Figure 5). The explored Hamiltonians refer to a family of 1‑D nearest‑neighbor Heisenberg. TITAN is trained with the HEA ansatz and tested on the SEL and SU2 ansätze. Gaussian initialization is adopted. The definition of $\Delta E$ follows the same meaning in Table R1.
>
> | Testing Ansatz | N=5 L=5| N=5 L=15 |N=10 L=10| N=15 L=5 | N=15 L=15 |
> |---------------------|--------|------------|---------|----------|----------|
> | SEL (TITAN Freezing)| +0.441 $\pm$ 0.054 | -0.081 $\pm$ 0.002 | -0.159 $\pm$ 0.374 | -0.671 $\pm$ 1.173 | -0.521 $\pm$  0.229|
> | SEL (Random Freezing) | +0.689 $\pm$ 0.269 | +2.280 $\pm$ 1.261 | +0.650 $\pm$ 0.178 | -0.055 $\pm$ 0.715 | +0.479 $\pm$  0.715|
> | SU2 (TITAN Freezing) | +0.123 $\pm$ 0.368 | -0.096 $\pm$ 0.301 | +0.119 $\pm$ 0.215 | -0.311 $\pm$ 1.714 | -0.301 $\pm$ 0.035|
> | SU2 (Random Freezing) | +1.626 $\pm$ 0.701 | -0.025 $\pm$ 0.290 | +0.262 $\pm$ 0.200 | +1.658 $\pm$ 13.714 | +0.389 $\pm$ 0.013|
>
> **Q1: Are shallow layers more prone to parameter freezing? Is this related to the gradient vanishing problem of classic NN?**
>
> **Reply**: For clarity, we address the reviewer's two concerns in turn in this response.
>
> $\bullet$ *Phenomenon of frozen parameters*. We would like to emphasize that the distribution of frozen parameters is **problem-dependent**. While Figure 4 shows that frozen parameters tend to concentrate in the shallow layers when applying HEA to isotropic Hamiltonians, **this behavior is not universal**.  As shown in Figures 6, 7, 9, 12, 16, and 17 of the Appendix, the distribution of frozen parameters **varies significantly** across different ansatzes and problem Hamiltonians.
>
> Due to the problem-dependent nature of parameter freezing, we propose using a deep learning model to learn and capture this behavior. As demonstrated in Section 4 and the corresponding Appendix figures, TITAN generalizes well across diverse settings,  significantly reducing measurement costs while preserving the final ground-state energy.
>
> $\bullet$ *Relation with vanishing gradients in classical neural networks*. TITAN is conceptually related to certain phenomena in classical NN, but *not to vanishing gradients*. In classical NN, vanishing gradients refer to the issue where shallow layers receive very small gradients, making them hard to train. In contrast, the frozen parameters identified by TITAN, as evidenced in Figures 6, 7, 9, and 17 of the Appendix, do not consistently appear in shallow layers and are not caused by gradient flow issues.
>
> A more relevant analogy is **neural network pruning**. In classical NN, some neurons may become inactive depending on the task or dataset. This is similar to our observation that certain VQE parameters remain inactive (i.e., frozen) depending on the Hamiltonian and ansatz. This connection may inspire further research into sparsity and parameter selection across classical and quantum learning models.
>
> [1] Hochreiter, Sepp. "The vanishing gradient problem during learning recurrent neural nets and problem solutions."
>
> [2] Pascanu R, Mikolov T, Bengio Y. On the difficulty of training recurrent neural networks International conference on machine learning. Pmlr, 2013: 1310-1318.

---

> > ### Comment · Reviewer_dadQ · 2025-08-07
> >
> > I would like to thank the authors for the detailed and clear response. My concerns are all resolved. The paper conveys more information than what the page-limit permits. My score remains the same.

---

> ### Author Response · Authors · 2025-08-07
> **Thank You and Follow Up to Reviewer dadQ**
>
> Dear Reviewer dadQ,
>
> We sincerely thank the reviewer for the positive feedback and for confirming that all earlier concerns have been fully addressed!
>
> We acknowledge the comment regarding information density relative to the page limit, and we will streamline the main text in the camera-ready version by relocating extended tables and explanations to the appendix.
>
> These edits will preserve clarity while ensuring the core contributions remain easily accessible.
> We appreciate the reviewer’s time and constructive input!

---

### Note · Authors · 2025-08-12

We sincerely thank all reviewers for their careful reading, constructive feedback, and positive evaluations of our work. Your comments have addressed the methodology, experimental design, presentation, and potential extensions from multiple perspectives, and we greatly appreciate the time and effort invested in the review process. After providing additional experiments and clarifications, we are pleased that all concerns were resolved and final scores were updated positively.

Once again, we sincerely thank all reviewers, ACs, SACs, and PCs for their engagement, patience, and valuable support throughout the review process!

---

### Decision · Program_Chairs · 2025-09-17

**Decision:**

Accept (poster)

**Comment:**

The paper reveals an insightful phenomenon: in variational quantum circuits, many parameters exhibit limited dynamics and can be frozen once optimized. Building on this, it proposes a learning-based framework to predict which parameters can be frozen and to what values, serving as initialization for variational quantum circuits.
The proposed method, TITAN, significantly speeds up convergence and reduces circuit evaluations while maintaining accuracy. The methodology and experimental results demonstrate a meaningful contribution. The rebuttal further clarifies improvements in performance and generalization capability, effectively addressing reviewers' concerns.